# Epidemiological trends in alcoholic cardiomyopathy burden: A 32-year global and Chinese analysis (1990–2021) with projections to 2036

Fei Yan[1,2,3◉], Changfen Wang[3◉], Jiulin Chen[3], Zhaoxing Cao[1,2], Runze Huang[1,2]*, Zhangrong Chen[1,2]*

1 Department of Cardiovascular Medicine, The Affiliated Hospital of Guizhou Medical University, Guiyang, Guizhou, China, 2 The Key Laboratory of Myocardial Remodeling Research, The Affiliated Hospital of Guizhou Medical University, Guiyang, Guizhou, China, 3 Department of Cardiovascular Medicine, People's Hospital of Qianxinan Prefecture, Xingyi, Guizhou, China

◉ These authors contributed equally to this work.
* runzehuang@pku.org.cn (RH); chenzhangrong71@163.com (ZC)

## Abstract

**Background:** Alcoholic cardiomyopathy (ACM), a preventable yet overlooked cardiomyopathy subtype, disproportionately affects chronic alcohol users through alcohol-induced myocardial damage. Characterized by delayed clinical onset, ACM typically manifests as irreversible heart failure in middle age (45-69 years), creating missed opportunities for early intervention. Despite global cardiovascular mortality declines, ACM burden continues rising paradoxically in certain middle-high income countries like China, revealing gaps in current prevention strategies. This study systematically evaluates ACM's global burden through prevalence, mortality and disability-adjusted life years (DALYs) metrics, using WHO-standardized methods to characterize spatiotemporal patterns across global, national, and subnational strata.

**Methods:** We utilized the Global Burden of Diseases, Injuries, and Risk Factors Study (GBD) 2021 methodological framework to estimate the age-standardized rates (ASRs) of prevalenceand, mortality and DALYs for ACM. These estimates were stratified across key demographic dimensions including: Age groups (15-95+ years), Sex, Geographical regions (21 GBD-defined regions), the Socio-Demographic Index (SDI) quintiles, 204 countries and territories during the observation period 1990-2021. Furthermore, we employed Bayesian Age-Period-Cohort (BAPC) modeling with integrated nested Laplace approximations to project disease burden trajectories through 2036, incorporating uncertainty quantification via Markov chain Monte Carlo simulations.

**Results:** Globally, ACM burden showed significant declines from 1990 to 2021, with age-standardized rates decreasing by 22.5-37.1% across prevalence, mortality and disability measures. However, China experienced a 200.4% case increase during this period, with rising mortality and disability rates contrary to global trends. The

**Data availability statement:** This study exclusively utilized de-identified aggregate data from the Global Burden of Disease Study 2021 (GBD 2021), which is permanently hosted and publicly accessible at zero cost through the IHME data portal: https://ghdx.healthdata.org/gbd-results-tool. No original datasets were generated in this work, as all analyzed data are openly available at the above link per IHME's data access policy.

**Funding:** ZChen No. 81960085 The National Natural Science Foundation of China gyfybsky-2022-44 Guizhou Medical University Doctoral Research Initiation Fund No. qiankehejichu-ZK[2023]yiban372 The Science and Technology Fund of Guizhou Provincial.

**Competing interests:** The authors have declared that no competing interests exist.

disease disproportionately affected males and middle-aged adults (45-69 years), with pronounced regional disparities in middle-high SDI areas. While population growth primarily drove disability-adjusted life year (DALY) increases, these regions also showed greatest potential for burden reduction through targeted interventions. Projections suggest continued global declines but only modest improvements in China through 2036.

**Conclusion:** In 2021, ACM remained a significant global health burden, particularly affecting middle-aged and elderly populations with distinct demographic disparities. This study identified three critical issues: China's unique epidemiological patterns, optimal intervention timing for high-risk populations, and sex-specific pathogenic mechanisms. Future research should prioritize developing precision prevention strategies for high-burden regions, including population-based alcohol control policies, early screening programs (especially for males aged 45-69), and personalized secondary prevention measures. Middle-high SDI regions warrant particular attention as priority intervention areas requiring cost-effectiveness implementation studies.

## Introduction

Alcoholic cardiomyopathy (ACM) is a progressive myocardial disorder caused by chronic excessive alcohol consumption and has become a major etiological contributor to heart failure [1]. Its pathogenesis involves multifactorial interactions, including direct alcohol toxicity to cardiomyocytes, oxidative stress, inflammatory activation, and neurohormonal dysregulation [2]. While traditional views attributed alcohol-related diseases like alcoholic fatty liver disease solely to drinking behavior, emerging evidence highlights the critical role of genetic susceptibility. For instance, patatin-like phospholipase domain-containing protein 3 (PNPLA3) variants are significantly associated not only with hepatic disorders but also with diabetes, insulin resistance, metabolic syndrome, and related cardiovascular events. Similarly, $17\beta$-hydroxysteroid dehydrogenase 13 (HSD17B13) variants participate in regulating metabolic disturbances and cardiovascular injury [3]. Global 2019 data indicated an ACM prevalence of 8.51 per 100,000 (about 710,000 cases) [4], yet high-risk groups (e.g., alcoholics carrying risk genotypes) exhibited prevalence rates up to 40%. Rising case numbers linked to changing drinking patterns (particularly among young males) [5] and pronounced geographical disparities—with Eastern European countries bearing the heaviest burden ($> 15/100,000$) [6] while Asia-Pacific regions show lower current prevalence—suggest that ACM development results from interactions between environmental exposure and genetic background, ultimately triggering cardiomyocyte apoptosis, myocardial hypertrophy, and pathological ventricular remodeling that progress to heart failure and complications [7,8]. Consequently, ACM control strategies must simultaneously address individual variability (genetic predisposition), drinking patterns, and regional heterogeneity, necessitating evidence-based prevention policies and optimized resource allocation globally [9]. This study employs global burden of disease (GBD) 2021 data to quantify ACM's

epidemiological triad (prevalence, mortality, disability-adjusted life years), addressing three key questions: (1) How does ACM burden evolve temporo-spatially amid global alcohol consumption shifts? (2) Which demographic features (age/sex/Socio-demographic Index) identify high-risk subgroups? (3) How can prevention strategies be optimized for precision interventions—particularly in developing nations undergoing drinking norm transitions like China?

## Methods

### Data source

The GBD 2021 study provides comprehensive epidemiological estimates for 369 diseases across 204 countries and 21 regions, stratified by five Socio-Demographic Index (SDI) categories from 1990 to 2021. To ensure data validity, GBD implemented a triangulated validation protocol: (1) Cross-verification between source data and DISMOD-MR 2.1 models (discrepancy threshold <15%); (2) Uncertainty quantification through 1000 bootstrap iterations generating 95% UIs; (3) Expert review of outliers by Institute for Health Metrics and Evaluation (IHME) clinical panels. Age-standardized rates (ASRs) for prevalence, mortality, and disability-adjusted life years (DALYs) were calculated using the WHO standard population. Estimated annual percentage changes (EAPCs) were derived from log-linear regression models. Stratified analyses examined variations across geography, sex, calendar years, and age groups (15–95 years).

**Ethical compliance statement:** This study utilized publicly available, de-identified aggregate data from the GBD 2021, no individual patient data were accessed. In accordance with the Declaration of Helsinki provisions for secondary data analysis, ethical approval was not required. This study utilized exclusively **de-identified aggregate data** from the Global Burden of Disease Study 2021 (GBD 2021), publicly accessible through the Institute for Health Metrics and Evaluation (IHME) portal (http://ghdx.healthdata.org/). [†]Per Article 32 of the Declaration of Helsinki (2013) governing secondary analysis of anonymized public data, and in accordance with our Institutional Review Board's exemption criteria, ethics approval was waived since no individual patient data or identifiers were accessed.

### Definition

**ACM** [10]

(1) Heavy alcohol consumption (>80g/day) for at least 5 years.

(2) Left ventricular end-diastolic diameter>2 SD above normal and left ventricular ejection fraction (LVEF) <50%.

(3) Exclusion of other causes of DCM, including hypertensive, valvular,and ischaemic heart disease (Fig 1).

**SDI**

The SDI serves as a composite metric for evaluating the sociodevelopmental status of various countries and regions. It is derived by computing the geometric mean of three distinct indicators: per capita income, the total fertility rate for women below the age of 25, and the mean years of education for individuals aged 15 and older [11]. SDI scores range from 0 (lowest development) to 1 (highest development). Based on these scores, countries are classified into five development tiers: low SDI (0.000 - 0.455), low-middle SDI (0.456 - 0.608), middle SDI (0.609 - 0.690), high-middle SDI (0.691 - 0.805), and high SDI (0.806 - 1.000). Detailed country-level SDI classifications are available through the (IHME) at http://ghdx.healthdata.org/.

**DALYs**

DALYs are a standardized metric for quantifying disease burden, representing the total number of healthy life years lost due to premature mortality and non-fatal health conditions. This composite measure comprises two core components: 1. Years of Life Lost (YLLs) calculated by multiplying the number of deaths by the reference life expectancy at the age of death; 2. Years Lived with Disability (YLDs) determined by multiplying the number of incident/prevalent cases by a disability weight (0 = full health, 1 = death) and duration of the condition.One DALY equates to the loss of one year of full health. The formula is expressed as: DALY = YLL + YLD.

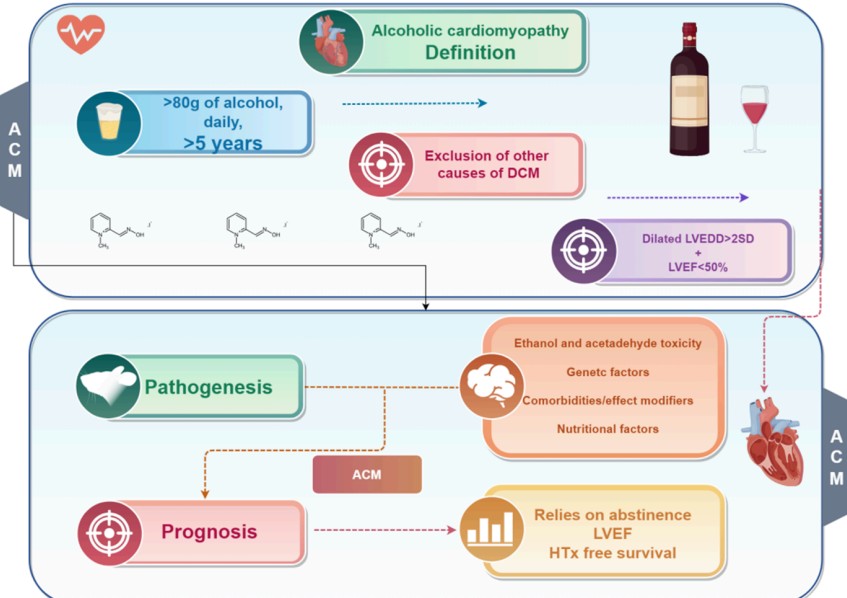

**Fig 1**. **Diagnostic criteria for Alcoholic cardiomyopathy.** By Figdraw.com.

## ASR

The ASR is a statistical metric that modifies the observed rates to accurately represent a population characterized by a uniform age distribution, enabling valid comparisons between populations with differing age distributions. This standardization process converts crude rates based on a population's actual age composition into adjusted rates using weights derived from a standard reference population's age distribution and the ASR (per 100,000 people) is calculated as follow:

$$ASR = \frac{\sum_{i=1}^{A} a_i w_i}{\sum_{i=1}^{A} w_i} \times 100\,000 \tag{1}$$

Among them, $a_i$ denotes the specific age ratio for the $i^{th}$ group, while $w_i$ represents either the population size or the assigned weight fot the $i^{th}$ group within the selected reference standard population.

## EAPC

The Estimated Annual Percentage Change (EAPC) is a statistical measure quantifying temporal trends in ASRs over a defined period. Calculated as:

$$EAPC = 100 \times [\exp(\beta) - 1]. \tag{2}$$

where $\beta$ represents the slope coefficient from a linear regression model of the natural logarithm of ASR against time, this metric provides both the magnitude and direction of rate changes. The associated 95% UI, derived from the regression model's confidence limits, enables trend classification:(1). Declining trend: EAPC and 95% UI upper bound < 0; (2). Rising trend: EAPC and 95% UI lower bound > 0.(3). Stable trend: 95% UI = 0. The absolute EAPC value reflects the rate of change magnitude, with statistical significance determined at p < 0.05 [12]. This method is particularly valuable for analyzing long-term disease incidence or mortality patterns in epidemiological studies.

## Methods and statistical analyses

This study comprehensively assessed ACM's global disease burden through descriptive epidemiology. We calculated crude and ASRs of prevalence, mortality, and DALYs. Stratified temporal trends (1990-2021) were visualized via sex-specific, regional, and national comparisons using multivariate techniques (heatmaps, time-series plots).Leveraging GBD 2021 data, we implemented:1.Joinpoint regression: Identified significant trend inflection points;2.Bayesian Age-Period-Cohort (BAPC) modeling: Age effects: Life-stage risk variations Period effects: Population-wide temporal shifts Cohort effects: Birth cohort lifetime risks (Computed via Integrated Nested Laplace Approximation (INLA), replacing MCMC to accelerate convergence from weeks to hours with <0.1% error); 3.Age-Period-Cohort ( APC ) analysis: Cohort effect estimation; 4.Stochastic frontier analysis: Healthcare efficiency assessment; 5.Concentration index: Health inequality quantification; All analyses used R 4.4.2. Bayesian posterior distributions yielded point estimates + 95% UIs; statistical significance required two-tailed p<0.05.

## Results

### 1. Burden profiles: China vs Global (1990–2021)

**1.1 Prevalence.** China demonstrated a sharply diverging trajectory from global patterns: age-standardized prevalence rates (ASPRs) tripled (+200.4%, 95% UI: 171.3 to 230.5) since 1990, reaching 1.6 per 100,000 (95% UI: 1.2 to 1.9) in 2021 and elevating China to the upper-medium global burden tier. Conversely, global ASPRs declined by 22.5% (95% UI: -26.6 to -18.2) to 6.2 per 100,000 (95% UI: 5.1 to 7.4). Extreme regional disparities were observed: Eastern Europe recorded the highest ASPRs (62.5 per 100,000, 95% UI: 51.9 to 75.0) versus near-absence in Andean Latin America (95% UI: 0 to 0.1) (Table 1; S1 Table). Critically, China's accelerated growth (EAPC = 4.79, 95% UI: 4.30 to 5.29) significantly deviated from the global downward trend (EAPC = –0.59, 95% UI: -0.83 to -0.35), indicating unique epidemiological drivers (Table 1; S1 Table)

**1.2 Mortality.** China revealed a disproportionate burden: the global age-standardized mortality rate (ASMR) declined by 37.1% (0.74 per 100,000) from 1990 to 2021 (95% UIs: -42.9 to -31.7 and 0.65 to 0.80), whereas China's ASMR spiked by 80.3% (0.096 per 100,000) (95% UIs: -43.2 to 261.7 and 0.017 to 0.152). Markedly divergent trajectories emerged: China's sustained ASMR increase (EAPC = 2.67) sharply contrasted with the global decline (EAPC = -1.72), underscoring population-specific risk determinants (Table 1).

**1.3 DALYs.** Amplified gender-regional disparities emerged: Global DALYs decreased by 29.3% (ASDR 25.3 per 100,000) (95% UIs: –35.3 to –23.7 and 22.4 to 27.5), whereas China experienced a 90.2% surge (0.78 per 100,000) (95% UIs: –34.4 to 274.9 and 0.15 to 1.24). Temporally, China's sustained ASDR increase (EAPC = 2.91, 95% UI: 2.59 to 3.22) demonstrated fundamental departure from the global decline (EAPC = –1.44, 95% UI: –2.61 to –0.27), signaling emergent public health challenges across regions (Table 1).

### 2. Age-specific burden of ACM: China vs Global population (2021)

Fig 2 and S2 Table reveal distinct epidemiological patterns: Globally, ACM predominantly affected individuals ≥ 15years, with male prevalence peaking at 70–74 years (peak case count at 65–69 years) and female rates at 65–69 years. Prevalence rose in 15–74 and 80–94-year-olds but declined among 75–84-year-olds and those ≥ 95 years.China manifested divergent patterns characterized by:(1) Persistent male dominance across all ages (male-to-female ratio: 7.8:1 through age 94);(2) Bimodal prevalence distribution in males (peaks at 15–49 and 60–69 years, with intervening decline at 50–64 years);(3) Steady female prevalence increase across all age groups. Critically, both case numbers and prevalence rates peaked earlier in China (45–49 years) than globally, implying accelerated disease onset potentially associated with: Occupational drinking cultures among prime working-age males Earlier initiation of alcohol exposure Sex-differential biological susceptibility (discussed in Sect 3) Mortality and DALY distributions appear in S1 Fig-S2 Fig and S2 Table, collectively supporting these sex-specific patterns.

**Table 1. Prevalent cases, deaths, and DALYs for ACM in 2021, and percentage change in ASRs per 100 000, by GBD region, from 1990 to 2021 (generated from data available at https://ghdx.healthdata.org/gbd-results-tool).**

| Location | Prevalence(95%UI) | | | | Deaths(95%UI) | | | | DALYs(95%UI) | | | |
|---|---|---|---|---|---|---|---|---|---|---|---|---|
| | 2021 | | Percentage change in ASRs from 1990to2021 | 1990-2021 | 2021 | | Percentage change in ASRs from 1990to2021 | 1990-2021 | 2021 | | Percentage change in ASRs from 1990to2021 | 1990-2021 |
| | No, in (95%UI) | ASRs per 100000 (95%UI) | | EAPC No, in (95%UI) | No, in (95%UI) | ASRs per 100000 (95%UI) | | EAPC No, in (95%UI) | No, in (95%UI) | ASRs per 100000 (95%UI) | | EAPC No, in (95%UI) |
| Global | 528429 (439582.3 to 639167.4) | 6.2 (5.1 to 7.4) | -22.5 (-26.6 to -18.2) | -0.59 (-0.83 to -0.35) | 64011.4 (56292.6 to 69518.7) | 0.741 (0.652 to 0.805) | -37.1 (-42.9 to -31.7) | -1.72 (-2.74 to -0.7) | 2185527.6 (1928589.3 to2368735.7) | 25.3 (22.4 to27.5) | -29.3 (-35.3 to-23.7) | -1.44 (-2.61 to-0.27) |
| China | 28103.4 (22175.4 to 34990.7) | 1.6 (1.2 to 1.9) | 200.4 (171.3 to 230.5) | 4.79 (4.3 to 5.29) | 1861 (317.6 to 2986.8) | 0.096 (0.017 to 0.152) | 80.3 (-43.2 to 261.7) | 2.67 (2.34 to 3) | 64745.9 (12595.5 to102876.1) | 3.4 (0.7 to5.4) | 90.2 (-34.4 to274.9) | 2.91(2.59 to3.22) |
| High-income Asia Pacific | 13148.4 (10795.1 to 16170.6) | 5 (4 to 6.1) | -25.8 (-32.1 to -18.5) | -1.19 (-1.28 to -1.1) | 489.1 (442.6 to 532.4) | 0.13 (0.118 to 0.141) | -60.4 (-64.2 to -55.9) | -3.29 (-3.39 to -3.19) | 13652.5 (12385.9 to14806.5) | 4.5 (4.1 to4.8) | -61 (-64.7 to-56.6) | -3.33 (-3.43 to-3.23) |
| High-income North America | 89214.3 (74203.6 to 108043.5) | 17.4 (14.6 to 20.9) | -11.7 (-19.4 to -1.6) | -0.49 (-0.6 to -0.38) | 6014.8 (5590.1 to 6391.3) | 1.014 (0.947 to 1.078) | -28 (-35.4 to -20.3) | -1.22 (-1.43 to -1.02) | 177117.9 (166935.4 to187218) | 32.9 (31 to34.7) | -24.6 (-30.9 to-17.3) | -0.98 (-1.14 to-0.81) |
| Western Europe | 88260.6 (71202.7 to 108293.2) | 13.4 (11 to 16.1) | -2.4 (-9.8 to 6) | 0.53 (0.12 to 0.95) | 4939.3 (4446.4 to 5379.1) | 0.59 (0.536 to 0.638) | -61.5 (-66.9 to -56.3) | -3.3 (-3.54 to -3.07) | 127835.8 (116945.2 to137643.6) | 17.6 (16.2 to18.9) | -58.9 (-64.2 to-53.7) | -3.07 (-3.34 to-2.81) |
| Australasia | 11895.7 (9653 to 14308.5) | 26.6 (21.8 to 31.8) | 98.5 (76.7 to 123.2) | 2.2 (1.73 to 2.67) | 433.4 (401.7 to 468.7) | 0.885 (0.825 to 0.957) | -27.7 (-35.6 to -18.9) | -0.96 (-1.5 to -0.42) | 12459.2 (11570.5 to13526.5) | 28.2 (26.2 to30.6) | -20.7 (-28.8 to-10.7) | -0.64 (-1.16 to-0.12) |
| Andean Latin America | 27.7 (20.2 to 37.7) | 0 (0 to 0.1) | 11.1 (-5.1 to 29.4) | 1.41 (0.92 to 1.9) | 2.8 (0.8 to 4.5) | 0.005 (0.001 to 0.008) | -56.5 (-76.4 to -29.5) | -2.04 (-2.6 to -1.48) | 49.7 (15.7 to80.4) | 0.1 (0 to0.1) | -55.4 (-74.6 to-30.1) | -1.87 (-2.45 to-1.29) |
| Tropical Latin America | 19237.2 (15573.6 to 23399) | 7.5 (6 to 9.1) | -37.1 (-42.2 to -31.9) | -2.13 (-2.35 to -1.92) | 1185.8 (1102.7 to 1269.7) | 0.449 (0.417 to 0.481) | -72.2 (-75.1 to -69) | -5.23 (-5.67 to -4.78) | 44412 (41533.2 to47434.8) | 16.8 (15.7 to18) | -70.8 (-73.8 to-67.6) | -5.09 (-5.55 to-4.63) |
| Central Latin America | 7975.7 (6594.5 to 9617.4) | 3 (2.5 to 3.6) | 12.7 (5.2 to 20.6) | 0.16 (0.06 to 0.27) | 478.4 (422 to 540.9) | 0.184 (0.162 to 0.207) | -16.8 (-29.1 to -4.4) | -1.31 (-1.65 to -0.98) | 17576.4 (15511.6 to19757.7) | 6.6 (5.8 to7.4) | -7.9 (-20.8 to5.1) | -0.9 (-1.2 to-0.6) |
| Southern Latin America | 2105.5 (1704.6 to 2586.9) | 2.7 (2.2 to 3.3) | -63.9 (-67 to -59.8) | -3.88 (-4.16 to -3.59) | 175 (155.5 to 194.6) | 0.206 (0.183 to 0.229) | -83.5 (-86.1 to -80.4) | -6.69 (-7.12 to-6.26) | 5024.7 (4484.6 to5557.2) | 6.1 (5.5 to6.8) | -83.9 (-86.2 to-81) | -6.72 (-7.14 to-6.3) |
| Caribbean | 9445.3 (7741.6 to 11510.4) | 18 (14.8 to 21.9) | 291.8 (259.7 to 324.1) | 5.65 (5.07 to 6.23) | 998.2 (799.8 to 1207.9) | 1.856 (1.483 to 2.256) | 225 (144 to 312.1) | 5.28 (4.69 to 5.87) | 30772.6 (24177.5 to37784.7) | 57.8 (45.2 to71.3) | 200.1 (130.1 to276.9) | 4.9(4.35 to5.46) |
| Central Europe | 42372.2 (35271.7 to 50685.5) | 25 (21.1 to 29.5) | 60.4(47 to 75) | 2.15 (1.95 to 2.34) | 5346.3 (4287.7 to 6113.3) | 2.668 (2.132 to 3.053) | -6.6 (-22.2 to 9.3) | 0.07 (-0.09 to 0.23) | 149199 (119377.9 to170872.8) | 81.7 (65.1 to93.8) | 5.9 (-11.7 to23.4) | 0.46 (0.24 to0.67) |

(Continued)

**Table 1.** (Continued)

| Location | Prevalence(95%UI) 2021 No, in (95%UI) | ASRs per 100000 (95%UI) | Percentage change in ASRs from 1990to2021 | 1990-2021 EAPC No, in (95%UI) | Deaths(95%UI) 2021 No, in (95%UI) | ASRs per 100000 (95%UI) | Percentage change in ASRs from 1990to2021 | 1990-2021 EAPC No, in (95%UI) | DALYs(95%UI) 2021 No, in (95%UI) | ASRs per 100000 (95%UI) | Percentage change in ASRs from 1990to2021 | 1990-2021 EAPC No, in (95%UI) |
|---|---|---|---|---|---|---|---|---|---|---|---|---|
| Eastern Europe | 173042.7 (142599.1 to 211308.7) | 62.5 (51.9 to 75) | 10.5 (3.3 to 17.8) | 0.5 (0.22 to 0.79) | 38834.7 (34484.1 to 42737) | 13.079 (11.624 to 14.372) | 40.9 (26.7 to 55.9) | 0.65 (-0.9 to 2.22) | 1426959.2 (1272059.1 to1567982.1) | 510.5 (456.7 to560.3) | 50.4 (34.6 to67.7) | 0.78 (-0.85 to2.43) |
| Central Asia | 5182 (4163.5 to 6386.5) | 5.4 (4.4 to 6.6) | 39 (28.5 to 48.8) | 1.21 (1 to 1.42) | 957.7 (788.7 to 1186.7) | 1.032 (0.856 to 1.272) | 33.9 (5.3 to 74.7) | 1.19 (0.51 to 1.88) | 35979.9 (29711.3 to44305.1) | 36.7 (30.4 to45.2) | 41.1 (11.8 to81.7) | 1.41 (0.74 to2.09) |
| North Africa and Middle East | 2524.4 (2072.5 to 3153.2) | 0.4 (0.4 to 0.5) | 5.5 (-1.2 to 13) | 0.24 (0.14 to 0.34) | 179.3 (51 to 350.6) | 0.04 (0.012 to 0.078) | -35.1 (-53.1 to -6.2) | -1.42 (-1.51 to -1.33) | 5724.4 (1684.9 to10424) | 1.1 (0.3 to2) | -36.5 (-53.4 to-11.6) | -1.52 (-1.61 to-1.42) |
| South Asia | 11199 (9043.9 to 13681.5) | 0.7 (0.5 to 0.8) | 4.2 (-2.8 to 12.4) | 0.22 (0.18 to 0.25) | 1450.7 (268.9 to 3395) | 0.094 (0.018 to 0.219) | -15.8 (-45.9 to 23.7) | -0.5 (-0.56 to -0.44) | 47908 (9929.7 to112144.6) | 2.9 (0.6 to6.7) | -18.2 (-45.2 to19.1) | -0.62 (-0.66 to-0.58) |
| Southeast Asia | 3219.2 (2644.3 to 3943) | 0.4 (0.4 to 0.5) | 22.9 (15.3 to 30.8) | 0.54 (0.46 to 0.62) | 365.1 (67.2 to 605.5) | 0.053 (0.01 to 0.085) | -10.1 (-42.8 to 30.6) | -0.6 (-0.74 to -0.47) | 13674.5 (2550.3 to21947.4) | 1.8 (0.3 to2.9) | -6.7 (-40 to32.6) | -0.44 (-0.55 to-0.33) |
| East Asia | 30311.5 (24088.8 to 37517.3) | 1.6 (1.3 to 2) | 158.1 (133 to 183.6) | 4.08 (3.68 to 4.48) | 1999.6 (426.3 to 3149.7) | 0.1 (0.022 to 0.156) | 56.2 (-45 to 181.5) | 2.07 (1.79 to 2.35) | 69749.7 (16481.7 to109076.5) | 3.6 (0.8 to5.6) | 71.5 (-33.5 to206.2) | 2.47 (2.18 to2.76) |
| Oceania | 21.1 (17 to 25.8) | 0.2 (0.2 to 0.3) | -12.9 (-19.7 to -6.4) | -0.62 (-0.69 to -0.56) | 5 (0.8 to 10.5) | 0.054 (0.008 to 0.118) | -28.5 (-55 to 14) | -1.22 (-1.31 to -1.12) | 206 (34.3 to419.9) | 1.9 (0.3 to3.9) | -26.8 (-56.7 to18.5) | -1.17 (-1.29 to-1.06) |
| Western Sub-Saharan Africa | 9298 (6788.4 to 12252.1) | 2.8 (2.1 to 3.6) | 6.4 (1.8 to 11.3) | 0.18 (0.05 to 0.31) | 147.9 (22.5 to 382.2) | 0.068 (0.012 to 0.169) | -60.8 (-73.4 to -37.6) | -3.64 (-3.93 to -3.35) | 5983.8 (1532.2 to14382.9) | 2.3 (0.6 to5.5) | -57.6 (-72.1 to-29.1) | -3.38 (-3.66 to-3.1) |
| Eastern Sub-Saharan Africa | 8012.5 (5774.5 to 10964) | 2.5 (1.9 to 3.4) | -3.2 (-8.1 to 2.8) | -0.21 (-0.26 to -0.17) | 2.9 (0.5 to 23.3) | 0.001 (0 to 0.01) | -28.2 (-61.9 to 3.8) | -1.23 (-1.34 to -1.12) | 860.7 (501.4 to1620.2) | 0.3 (0.2 to0.6) | -8.3 (-19.4 to2.5) | -0.41 (-0.45 to-0.36) |
| Central Sub-Saharan Africa | 1484.7 (1092.9 to 2035) | 1.6 (1.2 to 2.2) | -12.8 (-20.7 to -4) | -0.4 (-0.49 to -0.31) | 2 (0.3 to 12) | 0.003 (0.001 to 0.018) | -25 (-57.5 to 15.2) | -0.99 (-1.03 to -0.95) | 209.7 (115.2 to595.2) | 0.2 (0.1 to0.7) | -17.7 (-31.4 to2.1) | -0.63 (-0.69 to-0.57) |
| Southern Sub-Saharan Africa | 451.5 (292.9 to 643.1) | 0.6 (0.4 to 0.8) | -23.3 (-28.1 to -17.8) | -1.02 (-1.12 to -0.92) | 3.4 (0.7 to 9.2) | 0.009 (0.002 to 0.02) | -27.2 (-55.8 to 15.6) | -0.72 (-1.2 to -0.25) | 172.1 (72.5 to468.6) | 0.3 (0.1 to0.6) | -18.1 (-42.4 to29) | -0.65 (-0.95vto-0.34) |

95%UI=95% uncertainty intervals.

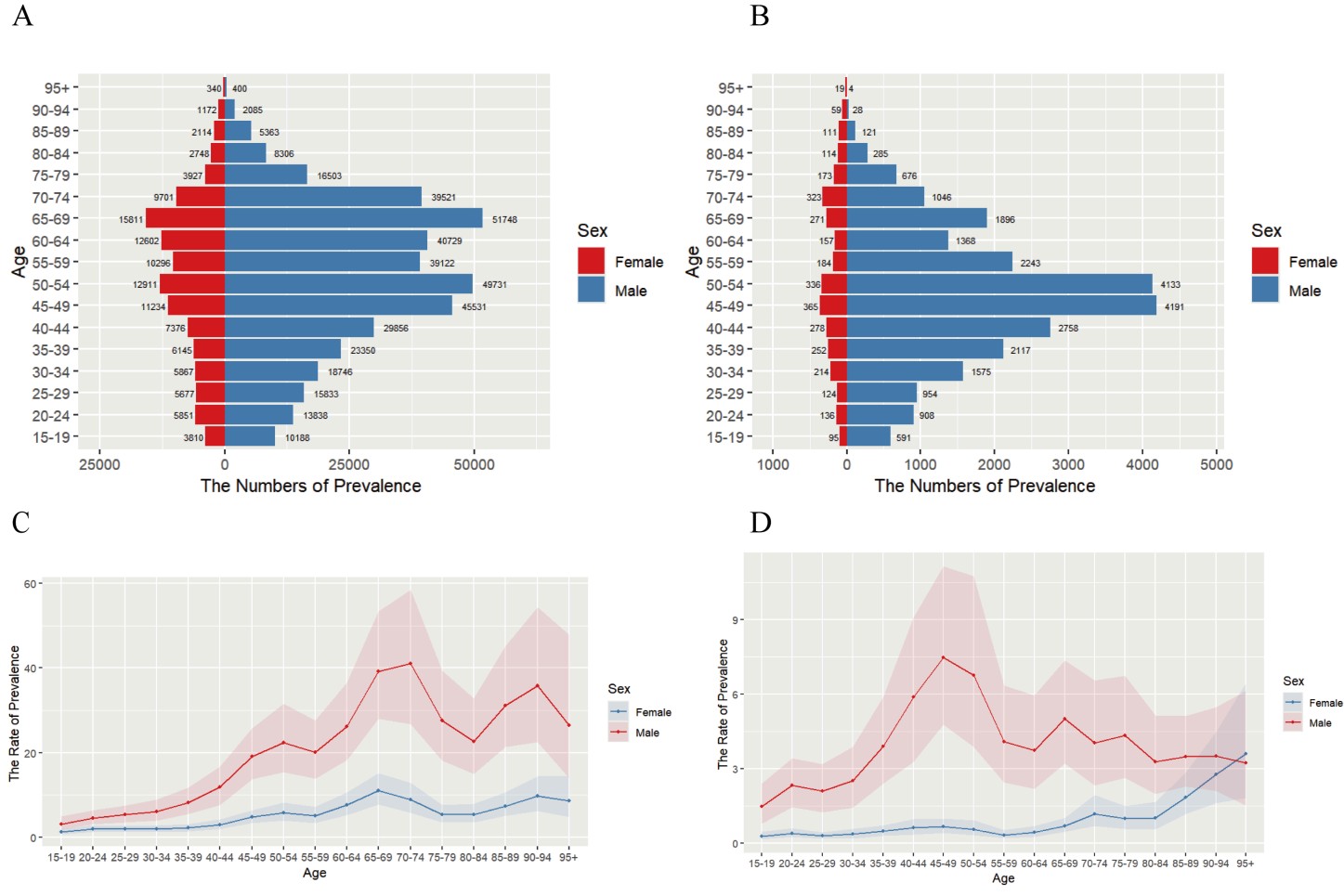

**Fig 2**. **Age-stratified case counts and crude prevalence rates of ACM by geographic scope: Global vs. China, 2021. A** Age-specific prevalence numbers for the global population; **B** Age-specific prevalence numbers for China; **C** Crude prevalence rate for the Global population; **D** Crude prevalence rate for China.

## 3. Temporal evolution of gender disparities (1990–2021)

The global gender gap peaked in 2006 (male ASPRs exceeded female by 186%) before narrowing (Fig 3A, S3 Table and S4 Table). China exhibited a divergent trajectory: male ASPRs accelerated post-2002, with absolute disparities persistently expanding despite slowed growth after 2015, reflecting intervention deficiencies targeting high-risk males (Fig 3B). Mortality disparities peaked in the middle-aged cohort (50–64 years), where Chinese male ASMRs reached 12.7-fold higher than females—significantly surpassing global patterns (Fig 3C, 3D). Close concordance between ASDR and mortality rates confirmed male-dominated burden as the primary prevention target (Fig 3E–3F).

## 4. Joinpoint regression analysis of the burden of ACM in China and the global population

Epidemiological Transition Distinct epidemic phases were identified ((Table 2); Figs 4–5 and S2 Fig): China exhibited two phases—ultrarapid expansion before 2004 (APC = 13.74, 95% UI: 12.94–14.55) followed by moderated but sustained growth post-2010 (APC = 4.15, 95% UI: 4.01 to 4.30). Globally, after a brief 1995–2004 ascent (APC = 0.85, 95% UI: 0.77 to 0.93), progressive decline prevailed, reflecting profound socio-demographic restructuring of disease burden.

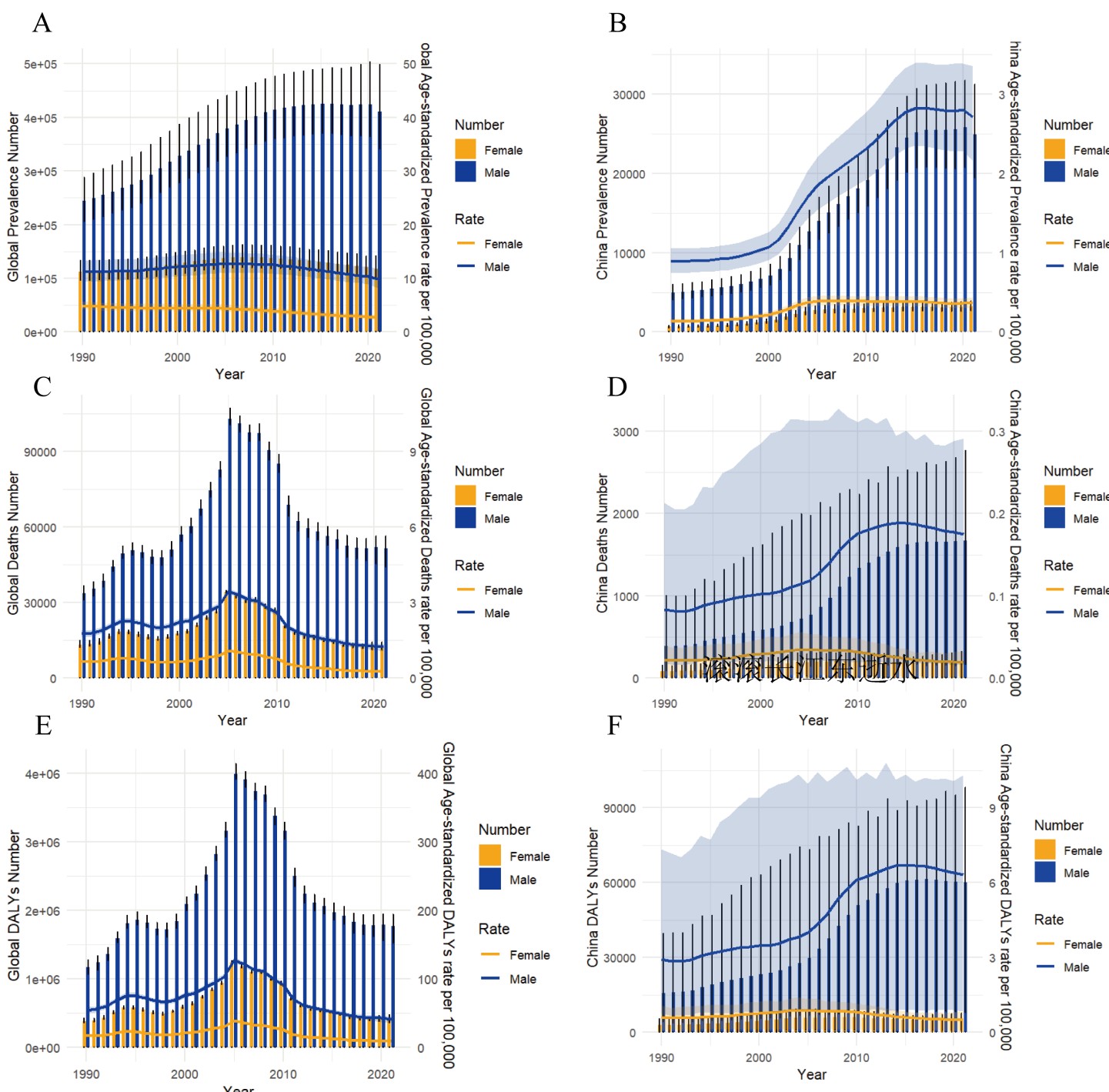

**Fig 3**. **Temporal trends in ACM burden (1990–2021): All-age case counts and age-standardized rates (prevalence/ASPRs, mortality/ASMRs, disability/ASDRs) stratified by sex. A** Global prevalence; **B** China prevalences; **C** Global mortality; **D** China mortality; **E** Global DALYs; **F** China DALYs.

**Table 2. Joinpoint regression analysis of temporal trends in age-standardized prevalence, mortality, and DALY rates (per 100,000 population) stratified by sex (both, male, female) and region (global vs. China), 1990–2021.**

| Gender | China_ASPR Period | APC(95%UI) | AAPC(95%UI) | China_ASDR Period | APC(95%UI) | AAPC(95%UI) | China_ASMR Period | APC(95%UI) | AAPC(95%UI) |
|---|---|---|---|---|---|---|---|---|---|
| Both | 1990-1995 | 0.74 (0.41 - 1.08) | 3.68 (3.54 - 3.82) | 1990-1992 | -1.43 (-3.24 - 0.42) | 2.11 (1.84 - 2.39) | 1990-1992 | -1.73 (-3.67 - 0.25) | 1.93 (1.71 - 2.15) |
| | 1995-2000 | 3.58 (3.11 - 4.04) | | 1992-1998 | 3.07 (2.64 - 3.51) | | 1992-1996 | 3.72 (2.73 - 4.72) | |
| | 2000-2004 | 13.74 (12.94 - 14.55) | | 1998-2001 | 1.00 (-1.05 - 3.09) | | 1996-2005 | 2.77 (2.54 - 3.01) | |
| | 2004-2014 | 4.15 (4.01 - 4.30) | | 2001-2005 | 3.35 (2.32 - 4.39) | | 2005-2010 | 6.50 (5.91 - 7.10) | |
| | 2014-2021 | -0.21 (-0.44 - 0.01) | | 2005-2010 | 7.37 (6.84 - 7.91) | | 2010-2014 | 0.49 (-0.26 - 1.24) | |
| | | | | 2010-2015 | 1.02 (0.62 - 1.43) | | 2014-2021 | -1.42 (-1.61 - -1.24) | |
| | | | | 2015-2021 | -1.21 (-1.42 - -1.00) | | | | |
| Male | 1990-1996 | 0.74 (0.46 - 1.02) | 3.71 (3.55 - 3.87) | 1990-1992 | -1.51 (-3.31 - 0.32) | 2.53 (2.26 - 2.80) | 1990-1992 | -1.34 (-3.15 - 0.50) | 2.44 (2.17 - 2.72) |
| | 1996-2000 | 3.43 (2.61 - 4.26) | | 1992-1996 | 3.73 (2.79 - 4.68) | | 1992-1997 | 3.80 (3.18 - 4.42) | |
| | 2000-2005 | 12.07 (11.50 - 12.65) | | 1996-2002 | 1.40 (0.94 - 1.85) | | 1997-2002 | 1.62 (0.96 - 2.28) | |
| | 2005-2015 | 4.32 (4.16 - 4.49) | | 2002-2005 | 4.03 (1.96 - 6.15) | | 2002-2005 | 3.91 (1.79 - 6.08) | |
| | 2015-2021 | -0.74 (-1.06 - -0.41) | | 2005-2010 | 9.04 (8.50 - 9.59) | | 2005-2010 | 8.43 (7.88 - 8.99) | |
| | | | | 2010-2015 | 1.83 (1.45 - 2.21) | | 2010-2014 | 1.71 (1.06 - 2.36) | |
| | | | | 2015-2021 | -1.10 (-1.30 - -0.91) | | 2014-2021 | -1.12 (-1.27 - -0.96) | |
| Female | 1990-1995 | 2.04 (1.50 - 2.59) | 3.24 (3.03 - 3.46) | 1990-1993 | -1.63 (-3.15 - -0.09) | -0.63 (-0.87 - -0.40) | 1990-1993 | -1.20 (-2.83 - 0.46) | -0.39 (-0.64 - -0.13) |
| | 1995-2000 | 8.20 (7.43 - 8.98) | | 1993-2004 | 3.99 (3.76 - 4.23) | | 1993-2004 | 4.68 (4.42 - 4.93) | |
| | 2000-2004 | 15.42 (14.12 - 16.73) | | 2004-2010 | -1.51 (-2.08 - -0.94) | | 2004-2010 | -1.40 (-2.04 - -0.76) | |
| | 2004-2013 | -0.07 (-0.31 - 0.18) | | 2010-2017 | -5.55 (-5.95 - -5.16) | | 2010-2017 | -5.74 (-6.18 - -5.30) | |
| | 2013-2021 | -0.91 (-1.17 - -0.66) | | 2017-2021 | -2.15 (-3.01 - -1.28) | | 2017-2021 | -2.18 (-3.12 - -1.23) | |

AAPC: average annual percent change presented for full period; APC: annual percent change; CI: confidence interval.

A

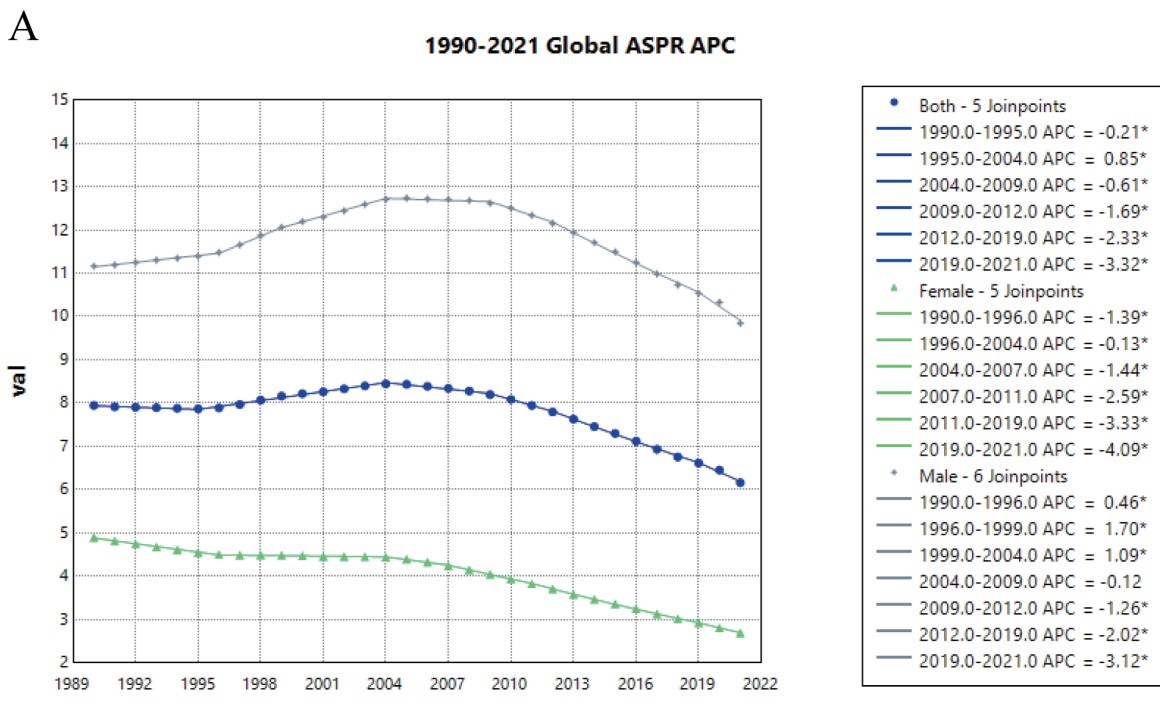

B

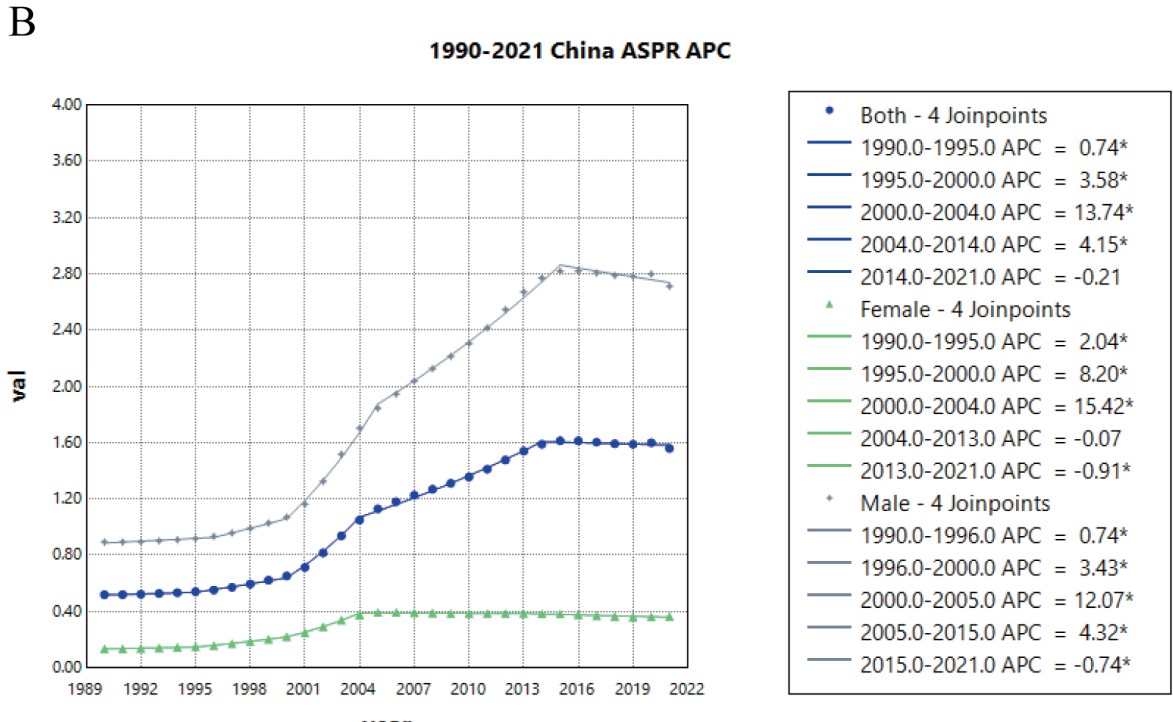

**Fig 4**. Joinpoint regression analysis of sex-stratified ASPRs for ACM: Global vs. China, 1990–2021. **A** Global; **B** China.

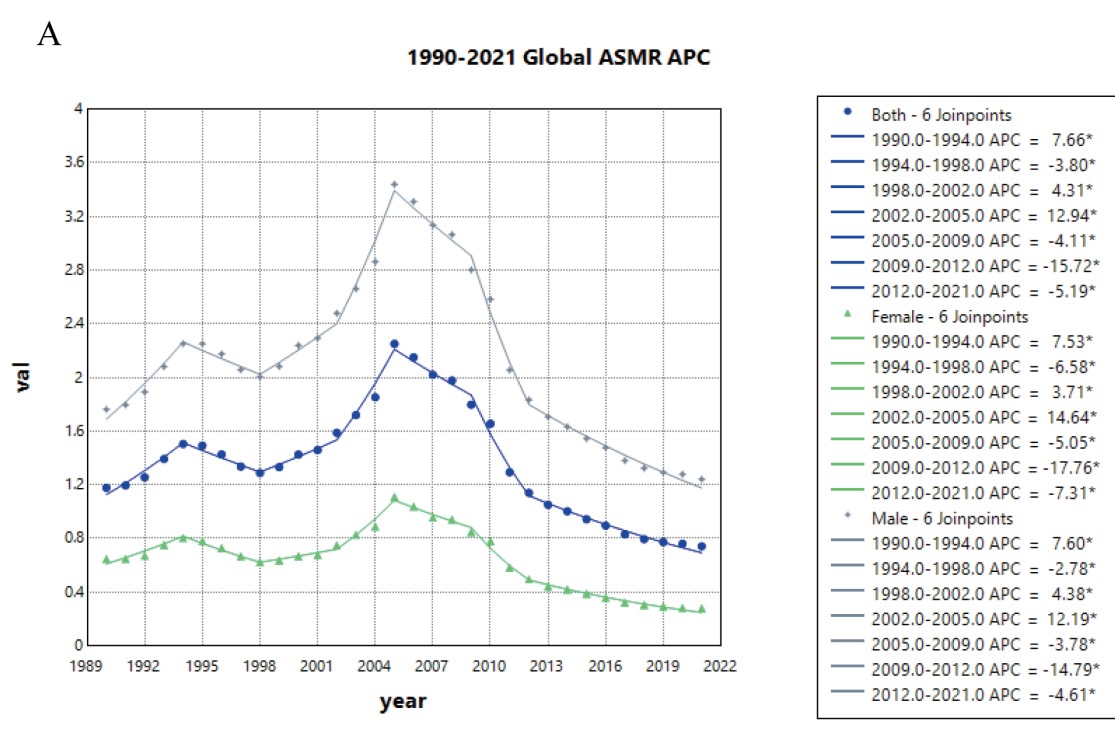

A

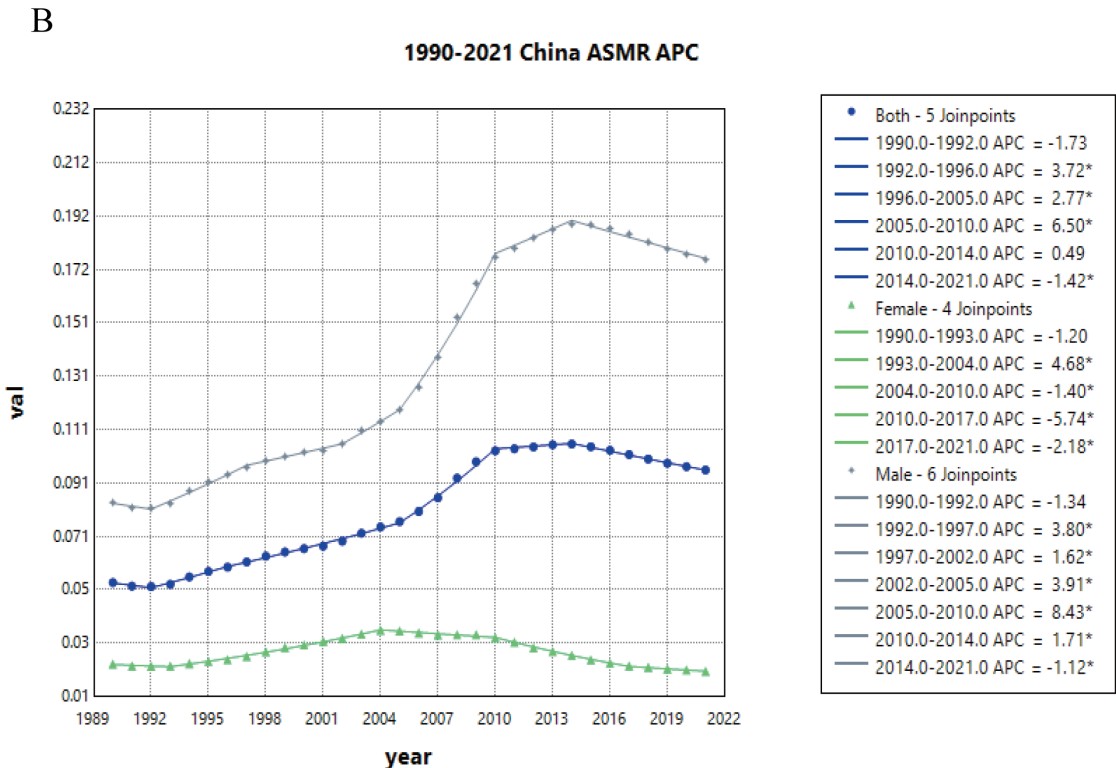

B

**Fig 5**. Joinpoint regression analysis of sex-stratified ASMRs for ACM: Global vs. China, 1990–2021. **A** Global; **B** China.

## 5. Burden of ACM across SDI quintiles in China and globally, 1990–2021

The burden of ACM, encompassing case numbers, mortality counts, ASPRs, and ASMRs, which exhibited marked regional disparities across SDI quintiles over the 1990-2021 observation period (Fig 6). High and high-middle SDI quintiles consistently demonstrated elevated case numbers and ASPRs compared to lower SDI groups throughout the study time-frame. While these metrics showed modest declines in recent decades, they remained substantially higher than baseline levels. Notably, China (categorized within the middle-high SDI group) achieved significantly greater ASPR reductions relative to both global averages and its SDI-matched counterparts, though minor resurgences emerged post-2010. Mortality patterns revealed distinct trajectories: high-middle SDI quintile exceeded high SDI areas in absolute mortality counts and ASMRs until plateauing at 2005 peaks. China's ASMR maintained stable low levels comparable to low, low-middle, and middle SDI strata throughout the study period.

## 6. Age-period-cohort patterns

Fig 7A and 7E depict the evolution of age-specific ACM prevalence globally and in China across seven survey cycles (1992, 1996, 2001, 2006, 2011, 2016, 2021). Globally, prevalence surged rapidly in the 0–50 age stratum, exhibiting

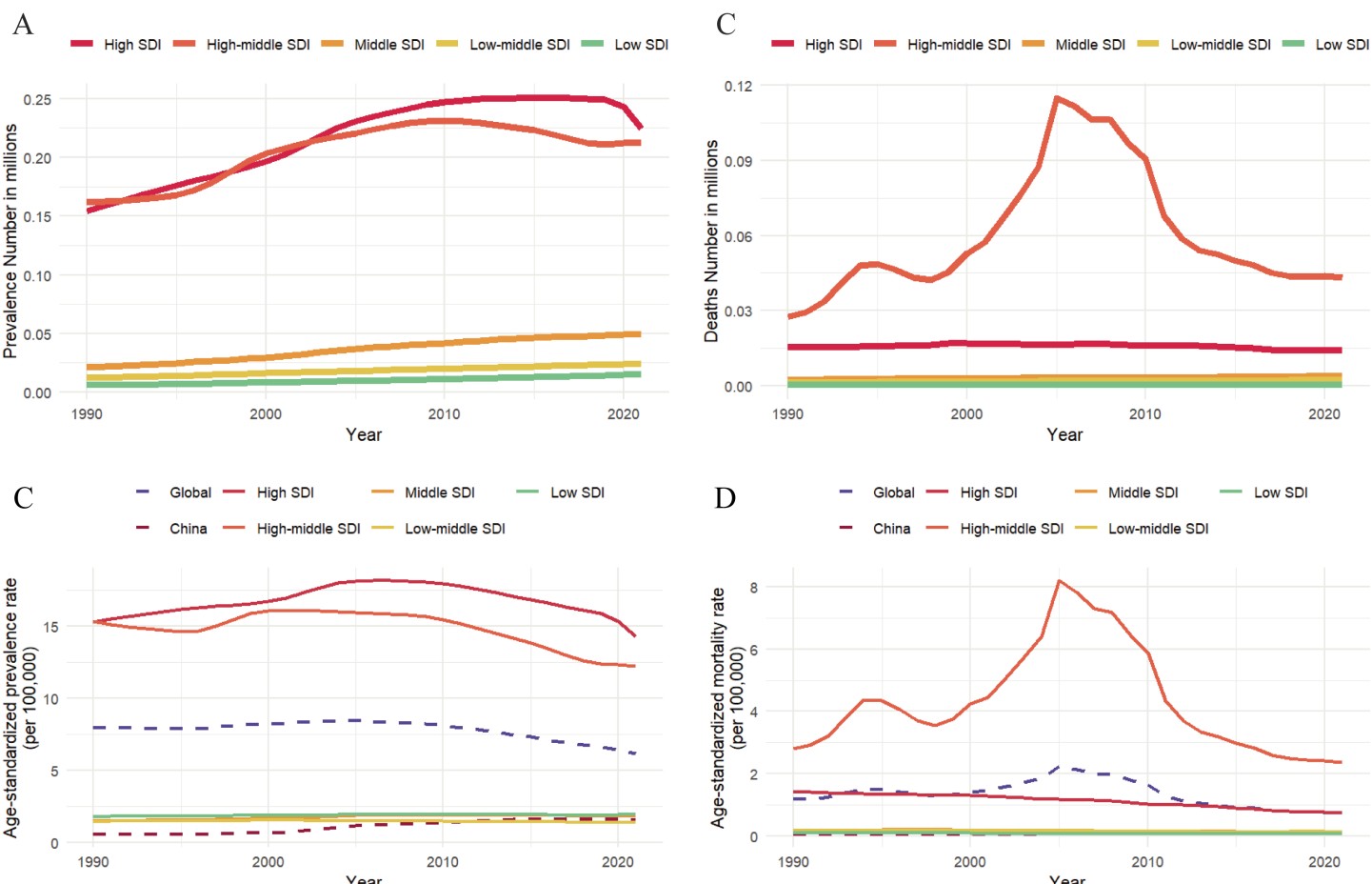

**Fig 6**. Joinpoint Temporal trends in ACM burden (1990–2021): prevalence/death counts and ASRs of prevalence and mortality stratified by SDI quintiles. **A** Prevalence numbers; **B** Mortality numbers; **C** ASPRs; **D** ASMRs.

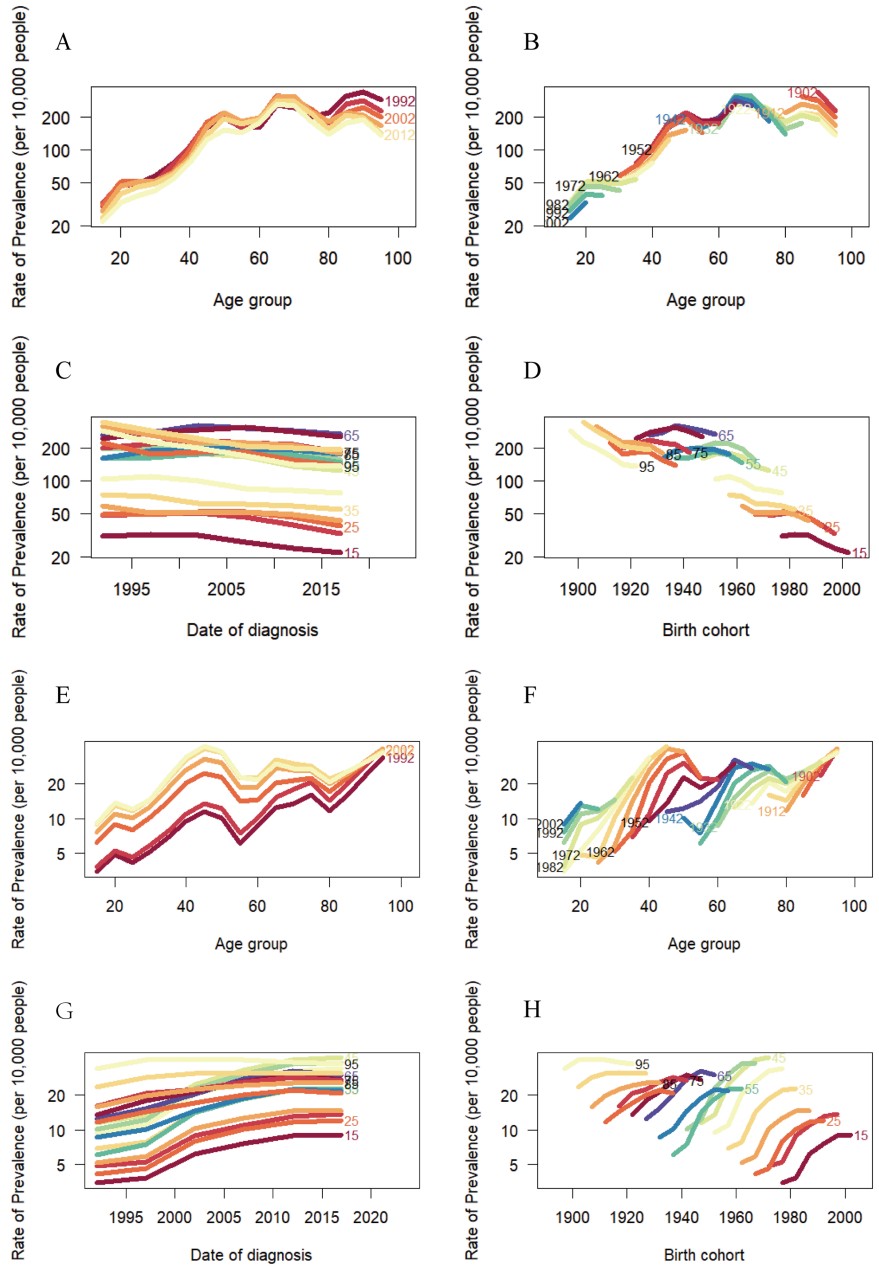

**Fig 7**. **Global and China's prevalence rates of ACM.** Global **A** and China's **E** age-specific prevalence rates of ACM according to time periods;Global **B** and China's **F** age-specific prevalence rates of ACM according to birth cohort; each line connects the age-specific prevalence for a 5-year cohort.Global **C** and China's **G** period-specific prevalence rates of ACM according to age groups; Global **D** and China's **H** The birth cohort-specific prevalence rates of ACM according to age groups; each line connects the birth cohort-specific incidence for a 5-year age group.

a characteristic triple-peak distribution at ages 70 and 90, with overall progressive aging shift (peak: 70→90 years). In stark contrast, China demonstrated unique early-onset epidemics concentrated at ages 45–49. Age-stratified prevalence dynamics are further delineated in Fig 7B and 7F. Longitudinal analysis (1990–2021) revealed divergent trajectories: gradual reductions across all age groups globally versus sustained increases in China, particularly pronounced among middle-aged (40–64 years) and elderly populations (≥ 65 years) (Fig 7C, 7G). Birth cohort patterns (Fig 7D, 7H) indicated

steady global prevalence escalation peaking at 65–70 years, contrasted by China's earlier apex (45–50 years) followed by decline.

## 7. Decomposition analysis of ACM disease burden: Comparative insights from China, worldwide, and SDI quintiles trends

We conducted a decomposition analysis on the DALYs data to assess the impacts of demographic aging, population increase, and epidemiological shifts on the burden of ACM between 1990 and 2021. Global DALYs increased significantly across all SDI quintiles except high SDI region, with parallel growth observed in China (Fig 8, S5 Table). Population expansion and aging accounted for 149.63% and 55.71% of global DALY increases respectively, whereas in China their contributions were more balanced (25.21% and 23.1%). Epidemiological changes exerted opposing effects: globally reducing DALY growth by 105.35% through risk factor mitigation, while amplifying China's burden by 51.64% due to residual disease susceptibility.The contributions of demographic and epidemiological drivers to DALYs exhibited marked quantitative disparities between China and global SDI strata, with population growth dominating globally versus balanced drivers in China (S5 Table).

## 8. Health inequity transition

The standardized inequality index (SII) for ASDR per 100,000 individuals shifted from 22.05 (95% uncertainty interval: 17.18 to 26.91) in 1990 to - 3.97 (95% uncertainty interval: - 11.92 to 3.97) in 2021, reflecting a fundamental reversal in the association between DALY rates and SDI from positive to negative correlation (Fig 9A). This period witnessed a socioeconomic reversal of ACM burden inequity, with ASDRs becoming disproportionately concentrated in lower-SDI nations. The Concentration Index (CI) of global DALYs approached equity neutrality (CI 1990 = - 0.59 vs CI 2021 = - 0.04, Fig 9B), demonstrating reduced cross-national health inequities in ACM burden. Temporal analysis revealed progressive SII convergence (Fig 9C): although regional-level disparities between low- and high-income countries diminished, persistent global inequities remained evident.

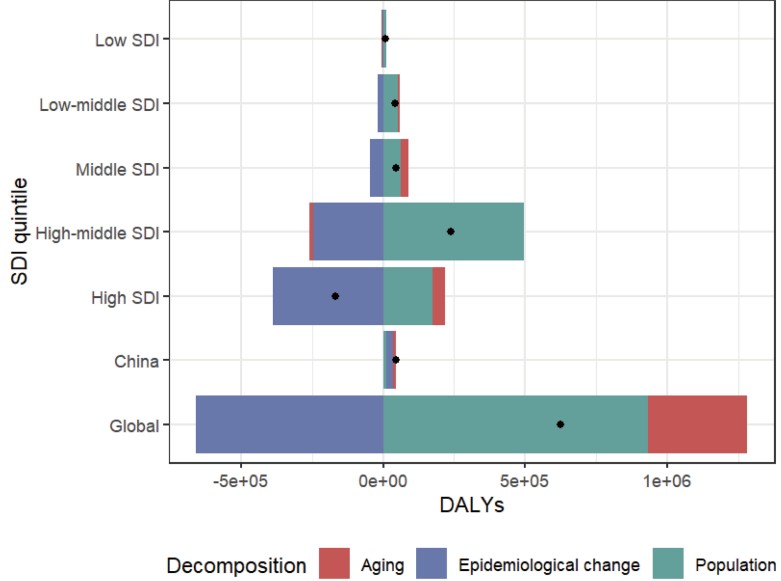

**Fig 8**. **Decomposition analysis of ACM DALYs changes (1990–2021) quantifies the contributions of population growth, aging, and epidemiological transition across geographic (global vs. China) and developmental strata.** The aggregate effect (black circular marker) reflects combined impacts of all three components, with positive values indicating drivers of DALYs increase and negative values representing reduction factors.

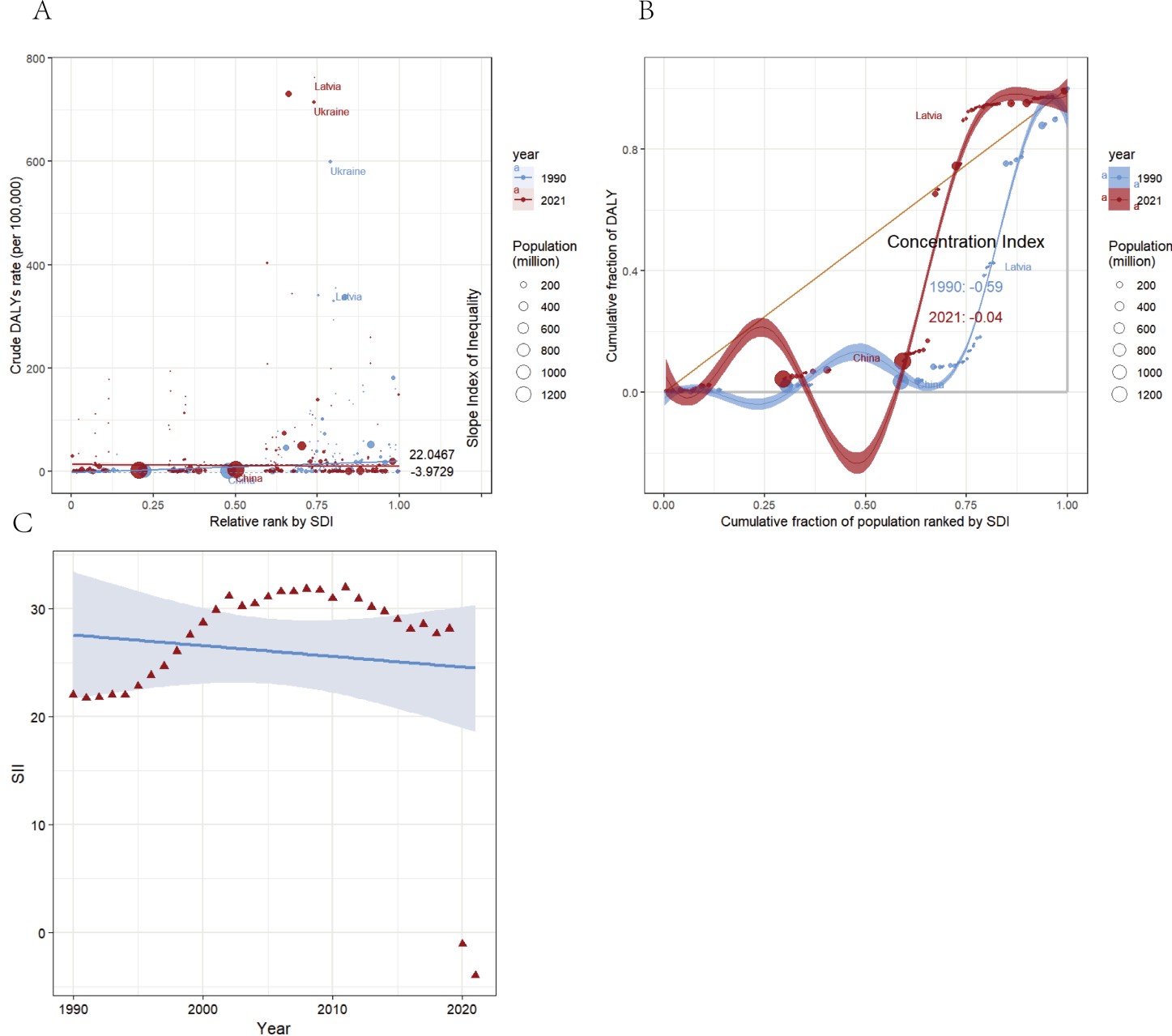

**Fig 9. Income-related health inequality in ACM Burden: Comparative analysis using Slope Index of Inequality (SII) and Concentration Index (1990 vs. 2021). A** Absolute income-related healthy inequality in ACM burden, presented using regression lines, 1990 vs. 2021. **B** Relative income-related healthy inequality in ACM burden, presented using concentration curves, 1990 vs. 2021. **C** Trendline demonstrates the trend in SII from 1990 to 2021.

## 9. Association between ASPRs and social development

Index across 21 distinct regions and 204 countries The association between SDI and ASPRs demonstrates a nonlinear asymmetric inverse-V pattern globally and across 21 regions (Fig 10A). High-middle and high SDI quintiles exhibited elevated ASPRs, Eastern Europe, categorized within the high SDI quintile, exhibited peak prevalence rates, while Andean

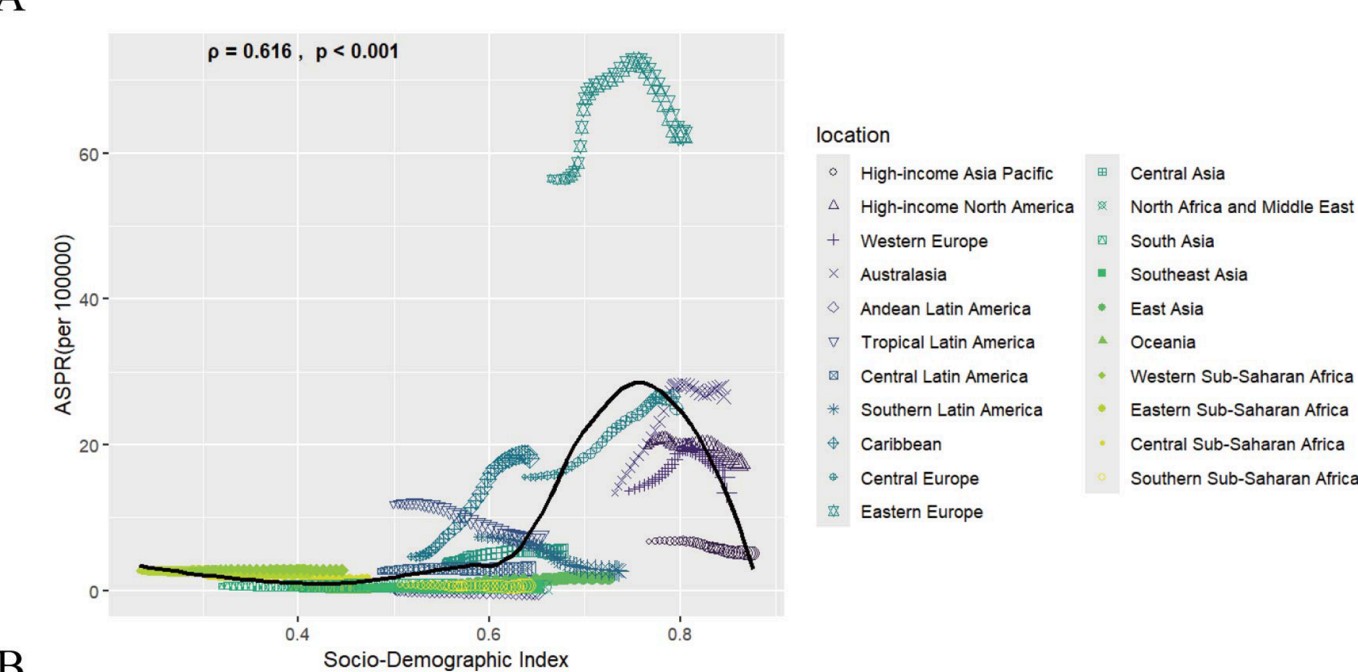

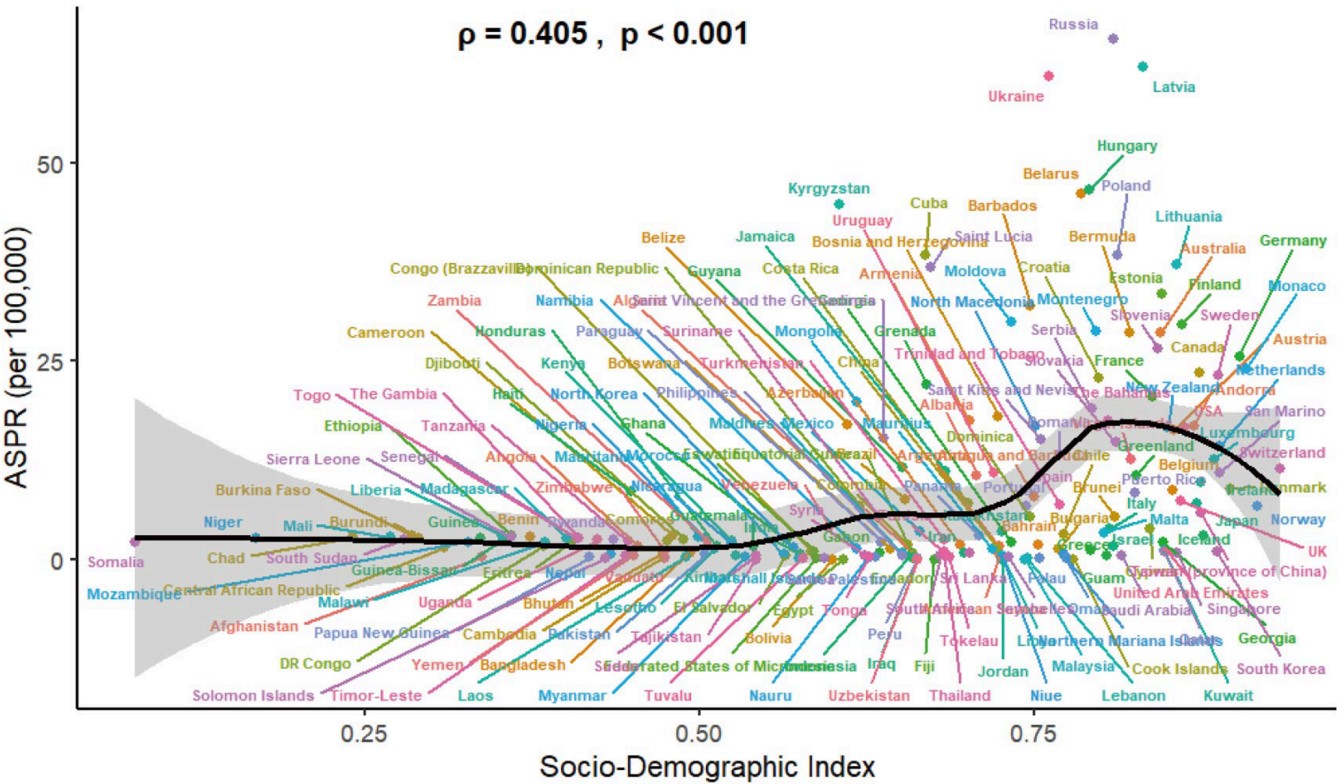

**Fig 10**. The association between ASPR and SDI at the regions(A) and nations (B) level.

Latin America (middle SDI quintile, SDI=0.652) demonstrated the lowest ASPRs. Cross-national analysis of 204 countries revealed high ASPRs concentration in high-middle and high SDI regions, with limited differentiation across lower SDI quintiles. Russia demonstrated the highest ASPRs, exceeding expected levels significantly alongside Latvia and Ukraine. Conversely, China's ASPR (high-middle SDI=0.72) remained substantially below projected values despite comparable SDI rankings (Fig 10B).

### 10. Frontier analysis of 204 nations

High-income nations underutilized healthcare advantages: Fig 11A demonstrates significant untapped health potential (avoidable DALY losses) during 1990–2021, while Fig 11B quantifies a 34% theory-reality gap in 2021 health attainment versus actual burden (S6 Table). Crucially, the gap widened disproportionately with national affluence, indicating suboptimal conversion of medical resources into burden reduction.

### 11. Projecting global and Chinese ACM burden (2022–2036): BAPC analysis.

Using GBD 1990–2021 data, we implemented the Bayesian APC-INLA model to project ASRs of ACM globally and in China. Results indicate consistent annual ASPR declines worldwide: males decreasing from 11 to 7 per 100,000 and females from 3 to 2 per 100,000 (Fig 12). In China, male ASPRs are projected to decline modestly (3→2 per 100,000), while female rates remain stable (0.383–0.398 per 100,000) (S7 Table). Parallel gradual reductions in ASDRs and ASMRs are anticipated globally and nationally during the projection period (Fig 12, S7 Table). Collectively, these forecasts suggest sustained 15-year declines in ASDRs (–0.9% annually), ASMRs (–1.2%), and mortality rates (–1.4%), though male disease burden persists substantially higher than females.

### Discussion

ACM represents a preventable form of non-ischemic cardiomyopathy whose epidemiological burden serves as a sentinel indicator of the complex interactions among cultural drinking norms, alcohol consumption behaviors, genetic predispositions, and public health governance. This study leveraged the Global Burden of Diseases, Injuries, and Risk Factors Study (GBD) 2021 dataset to systematically evaluate the trajectories of ACM-associated prevalence, mortality, and DALYs in China and globally during 1990–2021.

Data analysis revealed declining standardized rates (ASPR/ASMR/ASDR) alongside expanding absolute case counts globally, a paradoxical phenomenon primarily attributable to two concurrent drivers: (1) 32-year sustained public health efforts encompassing population education, health promotion, alcohol control, early diagnosis, and cardiac rehabilitation; (2) demographic aging and structural shifts—despite aggregate global improvement, significant inter-country disparities persist. SDI quintile analysis demonstrated marked geographical heterogeneity: upper-middle SDI regions exhibited substantially escalating absolute disease burden with parallel ASDR/ASMR increases, a pattern potentially presaging future burden trajectories in adjacent middle SDI regions, while middle/lower-middle/low SDI regions maintained relative stability. Classified as upper-middle SDI [13], China manifested counterintuitively robust upward trends: all 1990–2021 EAPCs registered high positive values (ASPR EAPC=4.79). Despite sustained lower ASPR/ASMR levels versus global and same-SDI-quintile peers with recent mild increases—indicating SDI is not the primary ACM burden determinant [14]—this derives from a tetrad of synergistic mechanisms: consumption transition (127% per capita alcohol increase 2000–2020) [15]; high-ethanol spirits dominance (>40% ethanol beverages linked to 53% cases) [16]; rural healthcare deficits (4.2% rehabilitation coverage); and genetic susceptibility (30.2% ALDH*2 mutation frequency in males) [17,18]. Country-specific burden analysis identified Eastern Europe as the most prominent among 21 global hotspots [19], with Ukraine/Latvia demonstrating the heaviest alcohol-attributable mortality burdens strongly correlated with the world's highest per capita consumption (Latvia >14L annually) [20]. Predominant spirits consumption coupled with prevalent heavy

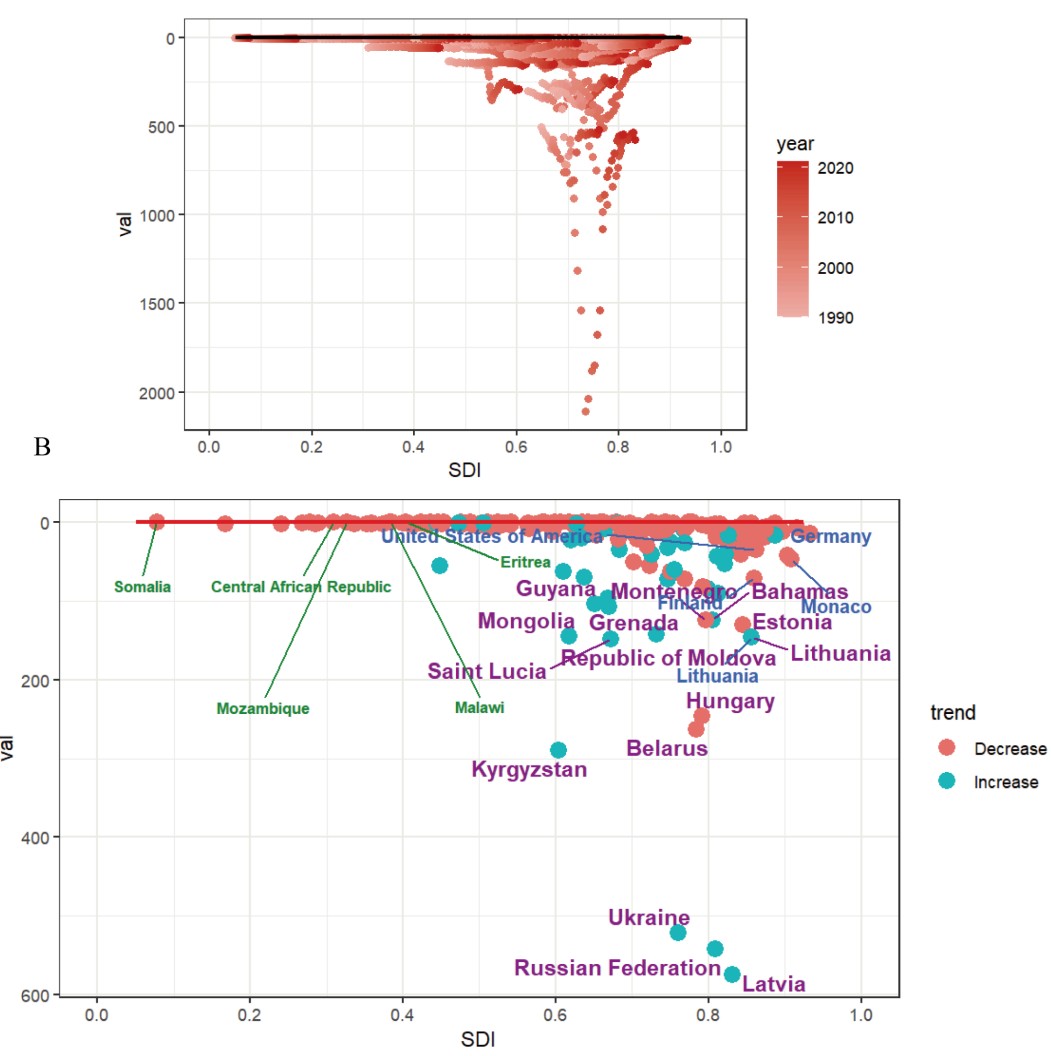

**Fig 11.** **Frontier analysis of ACM ASDRs by SDI in 2021.** **A** The frontier is shown in solid black, while countries and regions are represented by dots. **B** The top 15 countries with the largest effective difference (largest ACM DALYs gap from the frontier) are labeled in purple; examples of frontier countries with low SDI (< 0.5) and low effective difference are labeled in green (e.g., Somalia, Central African Republic, Mozambique, Malawi, Eritrea), and examples of countries and territories with high SDI (> 0.85) and relatively high effective difference for their level of development are labeled in blue (e.g., the United States, Lithuania, Monaco, Finland, Germany).Blue dots indicate an increase in age-standardized ACM DALYs rate from 1990 to 2021; Red dots indicate a decrease in age-standardized ACM DALYs rate between 1990 and 2021.

episodic drinking [21,22] established pronounced male dose-response relationships [23]. Although multiple Eastern European nations (e.g., Czech Republic/Estonia) implemented interventions like tax escalations, enforcement inconsistencies compromised efficacy [24].

Gender analysis indicates that the burden is significantly higher in males compared to females, likely stemming from differences in drinking patterns (occupational exposure/reinforcement of cultural norms) [25] and biological risks (metabolic pathways/genetic susceptibility) [26,27]. Cohort studies confirm that males have a markedly elevated risk of cardiac dysfunction (HR=1.75, 95% CI 1.60–1.92) [28] and fatal arrhythmias (HR=1.89, 95% CI 1.65–2.16) [29].

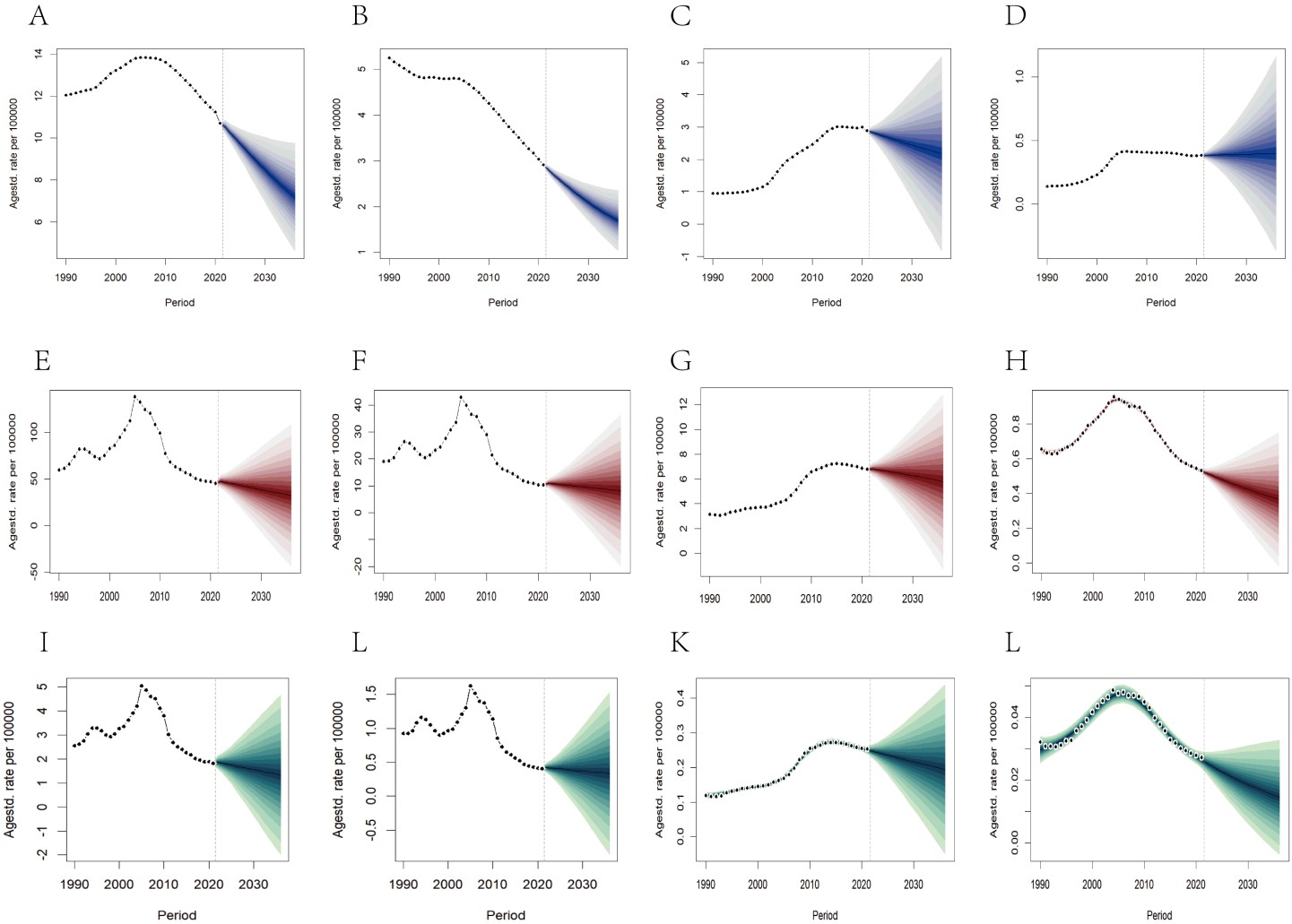

**Fig 12. Projection of ACM burden in China and global populations (2022-2036)**: Sex-specific Bayesian Age-Period-Cohort (BAPC) modeling analysis **(A)**; Global Females **(B)**; China males **(C)**; China Females **(D)**. Trends of ASDR from 1990 to 2036 in Global males **(E)**; Global Females **(F)**; China males **(G)**/China Females **(H)**. Trends of ASMR from 1990 to 2036 in Global males **(I)**; Global Females **(J)**; China males **(K)**; China Females **(L)**.

Meanwhile, age-related data reveal that China exhibits a distinct age stratification with a bimodal distribution, characterized by an initial peak at 45–50 years and a primary peak at 65–69 years. Based on the above analysis, policy development should prioritize a dual-track strategy: implementing immediate rigorous controls in middle-high SDI regions experiencing sharp burden surges, while proactively deploying targeted prevention strategies in middle SDI regions to curb potential burden escalation; concurrently, prevention and treatment frameworks must integrate gender and age stratification as core priorities, reinforcing two intervention windows: strengthening WHO alcohol control measures for <25 years (requiring blocking the 37% rise in youth digital marketing exposure) and optimizing screening for >45 years (addressing rural screening coverage gaps below 18%) [30]. Regarding fiscal and publicity dimensions: implement WHO progressive alcohol taxation (20–50% tax rates potentially reducing consumption by 16–28%) [16] while enhancing advertising regulations: restricting youth digital marketing aligned with the FCTC Convention and delivering precision education through gender/age-stratified interventions, enforcing restrictions benchmarked against the WHO Framework Convention

on Tobacco Control [31]; additionally, for countries with significant regional disease burden disparities, incorporate genetic susceptibility screening into routine practice. However, this study has methodological limitations: 1. Death misclassification risk: clinical diagnostic overlap between ACM and other cardiomyopathies (e.g., dilated cardiomyopathy), particularly in primary care with high misdiagnosis rates, potentially underestimating burden; 2. Alcohol exposure measurement bias: alcohol consumption estimates reliant on household surveys and tax data without correction for unrecorded consumption (e.g., homemade spirits); 3. Inadequate comorbidity control: the GBD model incompletely integrates synergistic effects of conditions like depression and liver disease, potentially distorting attribution analysis; 4. Missing policy variables: failure to quantify provincial/national alcohol policy strength (e.g., tax enforcement, advertising ban scope), constraining policy-burden association analysis; 5. Alcohol-type data fragmentation: no differentiation between spirits, imported liquors, beer etc. Future efforts should enhance model validity through deep learning of electronic medical records to correct misdiagnoses, integrate alcohol biomarker monitoring, and develop policy strength indices.

## Conclusion

ACM, a preventable condition through lifestyle modifications, demonstrates the critical importance of early clinical intervention for significantly improving patient prognosis. Global epidemiological data from 1990 to 2021 indicate relative progress in mitigating ACM burden across most regions. However, China's epidemiological trajectory presents a striking contrast, exhibiting a persistent upward trend during this period. Our analysis reveals pronounced disparities in ACM burden distribution across three key demographic dimensions: (1) absolute male predominance (male-to-female prevalence ratio of 3:1); (2) disproportionate concentration of disease burden among middle-aged (45-49 years) and elderly (65+ years) populations; and (3) geographic clustering predominantly in middle-high sociodemographic index (SDI) regions, with Eastern European countries demonstrating the highest standardized prevalence rates. This epidemiological divergence underscores the urgent need to address China's accelerating ACM burden, mandating policymakers to immediately implement three-tiered actions: First, enforce progressive spirits taxation (rates $\geq$ 30%) and comprehensively block youth digital marketing channels (curbing the current 37% surge in exposure rates); second, elevate rural echocardiography screening coverage to $\geq$ 50% by 2025 (current coverage gap <18%) while implementing compulsory occupational alcohol restrictions in high-exposure occupations such as manufacturing and transportation; third, incorporate an annual ACM mortality reduction rate $\geq$ 5% into local performance assessment systems, establishing real-time monitoring of tax evasion rates. Researchers must prioritize four critical domains: establishing clinical diagnostic thresholds for ACM-predisposing genetic variants (e.g., PNPLA3/HSD17B13 loci), developing rapid biomarker detection kits, completing multi-center RCTs on digital alcohol cessation therapies, and constructing agent-based modeling (ABM) simulations for alcohol taxation policies—ultimately forging an epidemiologically-tailored prevention-control framework to accelerate achievement of Sustainable Development Goal 3.4 targets.

## Supporting information

**S1 Fig. Global and China age-specific numbers and crude morality rates or of ACM in 2021. A** Global age-specific morality numbers. **B** China age-specific deaths numbers **C** Global crude morality rates. **D** China crude morality rates; Global and China age-specific numbers and crude DALY rates of ACM in 2021. **E** Global age-specific DALYs numbers. **F** China age-specific DALYs numbers . **G** Global crude DALY rates. **H** China crude DALY rates.
(PNG)

**S2 Fig. Joinpoint regression analysis of the sex-specific ASDRs for ACM in China and worldwide from 1990 to 2021. A** Global analysis; **B** China's analysis.
(PNG)

**S1 Table. The Burden of Alcoholic Cardiomyopathy (ACM) in 2021: Age-Standardized Rates (ASPR) and Estimated Annual Percentage Changes (EAPC) in China and Globally.**
(XLSX)

**S2 Table. Global and China age-specific numbers and crude rates of prevalence,morality and DALY for ACM in 2021.**
(XLSX)

**S3 Table. Global all-age cases and ASPRs, ASMRs and ASDRs of ACM by sex from 1990 to 2021.**
(XLSX)

**S4 Table. All-age cases and ASPRs, ASMRs and ASDRs of ACM in China by sex from 1990 to 2021.**
(XLSX)

**S5 Table. Changes in ACM DALYs according to population-level determinants of population, aging, and epidemiological change from 1990 to 2021 at the global , China and by SDI quintiles.**
(XLSX)

**S6 Table. The frontier analysis of ACM ASDRs by SDI in 2021.**
(XLSX)

**S7 Table. The frontier analysis of ACM ASDRs by SDI in 2021.** Global and China Predicted ASPRs,ASDRs and ASMRs by EAPC models.
(XLSX)

## Author contributions

**Conceptualization:** Changfen Wang.

**Data curation:** Fei Yan, Changfen Wang, Jiulin Chen, Zhaoxing Cao.

**Formal analysis:** Changfen Wang.

**Methodology:** Fei Yan, Changfen Wang, Jiulin Chen, zhangrong Chen.

**Visualization:** Fei Yan, Zhaoxing Cao.

**Writing – original draft:** Fei Yan, Changfen Wang.

**Writing – review & editing:** Changfen Wang, Runze Huang, zhangrong Chen.

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
