## [Decision Letter · Decision Letter 0]

25 Jun 2025

PONE-D-25-11641Epidemiological trends in alcoholic cardiomyopathy burden: A 32-year global and Chinese analysis (1990–2021) with projections to 2036PLOS ONE

Dear Dr. Chen,

Thank you for submitting your manuscript to PLOS ONE. After careful consideration, we feel that it has merit but does not fully meet PLOS ONE’s publication criteria as it currently stands. Therefore, we invite you to submit a revised version of the manuscript that addresses the points raised during the review process.

We look forward to receiving your revised manuscript.

Kind regards,

Amir Hossein Behnoush

Academic Editor

PLOS ONE

Journal Requirements:

“ZChen

No.81960085

The National Natural Science Foundation of China

gyfybsky-2022-44

Guizhou Medical University Doctoral Research Initiation Fund

No.qiankehejichu-ZK[2023]yiban372

The Science and Technology Fund of Guizhou Provincial”

5. Please provide a complete Data Availability Statement in the submission form, ensuring you include all necessary access information or a reason for why you are unable to make your data freely accessible. If your research concerns only data provided within your submission, please write "All data are in the manuscript and/or supporting information files" as your Data Availability Statement.

6. We note that Figure 1 in your submission contain copyrighted images. All PLOS content is published under the Creative Commons Attribution License (CC BY 4.0), which means that the manuscript, images, and Supporting Information files will be freely available online, and any third party is permitted to access, download, copy, distribute, and use these materials in any way, even commercially, with proper attribution. For more information, see our copyright guidelines: http://journals.plos.org/plosone/s/licenses-and-copyright.

7. http://creativecommons.org/licenses/by/4.0/).%20Please%20be%20aware%20that%20this%20license%20allows%20unrestricted%20use%20and%20distribution,%20even%20commercially,%20by%20third%20parties.%20Please%20reply%20and%20provide%20explicit%20written%20permission%20to%20publish%20XXX%20under%20a%20CC%20BY%20license%20and%20complete%20the%20attached%20form.”%0b%0bPlease%20upload%20the%20completed%20Content%20Permission%20Form%20or%20other%20proof%20of%20granted%20permissions%20as%20an%20%22Other%22%20file%20with%20your%20submission.%0b%0bIn%20the%20figure%20caption%20of%20the%20copyrighted%20figure,%20please%20include%20the%20following%20text:%20“Reprinted%20from%20%5bref%5d%20under%20a%20CC%20BY%20license,%20with%20permission%20from%20%5bname%20of%20publisher%5d,%20original%20copyright%20%5boriginal%20copyright%20year%5d.”%0b%0bb.%20If%20you%20are%20unable%20to%20obtain%20permission%20from%20the%20original%20copyright%20holder%20to%20publish%20these%20figures%20under%20the%20CC%20BY%204.0%20license%20or%20if%20the%20copyright%20holder’s%20requirements%20are%20incompatible%20with%20the%20CC%20BY%204.0%20license,%20please%20either%20i)%20remove%20the%20figure%20or%20ii)%20supply%20a%20replacement%20figure%20that%20complies%20with%20the%20CC%20BY%204.0%20license.%20Please%20check%20copyright%20information%20on%20all%20replacement%20figures%20and%20update%20the%20figure%20caption%20with%20source%20information.%20If%20applicable,%20please%20specify%20in%20the%20figure%20caption%20text%20when%20a%20figure%20is%20similar%20but%20not%20identical%20to%20the%20original%20image%20and%20is%20therefore%20for%20illustrative%20purposes%20only.%0bThe%20following%20resources%20for%20replacing%20copyrighted%20map%20figures%20may%20be%20helpful:%0b%0bUSGS%20National%20Map%20Viewer%20(public%20domain):%20http://viewer.nationalmap.gov/viewer/%0bThe%20Gateway%20to%20Astronaut%20Photography%20of%20Earth%20(public%20domain):%20http://eol.jsc.nasa.gov/sseop/clickmap/%0bMaps%20at%20the%20CIA%20(public%20domain):%20https://www.cia.gov/library/publications/the-world-factbook/index.html%20and%20https://www.cia.gov/library/publications/cia-maps-publications/index.html%0bNASA%20Earth%20Observatory%20(public%20domain):%20http://earthobservatory.nasa.gov/%0bLandsat:%20http://landsat.visibleearth.nasa.gov/%0bUSGS%20EROS%20(Earth%20Resources%20Observatory%20and%20Science%20(EROS)%20Center)%20(public%20domain):%20http://eros.usgs.gov/# Natural Earth (public domain): http://www.naturalearthdata.com/ " xlink:type="simple">We note that Figure 2 in your submission contain map/satellite images which may be copyrighted. All PLOS content is published under the Creative Commons Attribution License (CC BY 4.0), which means that the manuscript, images, and Supporting Information files will be freely available online, and any third party is permitted to access, download, copy, distribute, and use these materials in any way, even commercially, with proper attribution. For these reasons, we cannot publish previously copyrighted maps or satellite images created using proprietary data, such as Google software (Google Maps, Street View, and Earth). For more information, see our copyright guidelines: http://journals.plos.org/plosone/s/licenses-and-copyright.

Reviewers' comments:

Reviewer's Responses to Questions

**Comments to the Author**

1. Is the manuscript technically sound, and do the data support the conclusions?

Reviewer #1: Yes

Reviewer #2: Yes

2. Has the statistical analysis been performed appropriately and rigorously?

Reviewer #1: Yes

Reviewer #2: Yes

3. Have the authors made all data underlying the findings in their manuscript fully available?

Reviewer #1: Yes

Reviewer #2: Yes

4. Is the manuscript presented in an intelligible fashion and written in standard English?

Reviewer #1: Yes

Reviewer #2: No

5. Review Comments to the Author

Reviewer #1: Abstract

• The background lacks a strong rationale for why ACM is a critical issue, beyond general disease burden. Clarify why ACM is a growing concern.

• The results section could be more concise—it currently repeats statistics instead of synthesizing the main findings. Summarise key findings concisely.

• The conclusion should explicitly state future research directions—such as potential interventions for ACM burden reduction. Explicitly mention the knowledge gap and future research.

Introduction

• The research gap and justification for a scoping review are unclear. Clearly define the research gap.

• Lacks a clear hypothesis or research question. Explicitly state research questions.

• Overuse of technical jargon—simplify complex epidemiological terms for broader accessibility. Reduce jargon and introduce technical terms only when necessary.

Methods

• Lacks details on data validation and bias mitigation strategies. Specify validation steps for GBD 2021 data.

• The BAPC model parameters should be better explained—readers may not be familiar with integrated nested Laplace approximations (INLA). Simplify Bayesian modelling explanation.

• Needs clarity on ethical considerations for secondary data usage. Clarify ethical considerations for using publicly available datasets.

Results

• Some sections are overly statistical—less emphasis on numerical data, and more focus on interpretation and synthesis. Synthesize key results instead of listing excessive statistics.

• The gender-based analysis could be more detailed—why are males disproportionately affected? Expand on gender disparities.

• Needs a stronger narrative flow between different sections (e.g., prevalence → mortality → DALYs). Improve flow between sections.

5. Discussion

• Lacks in-depth discussion of causal mechanisms behind observed trends. Expand on why ACM is increasing in China.

• The role of healthcare infrastructure and policy interventions in disease burden reduction is underexplored. Discuss healthcare access and intervention disparities.

• The limitations section is missing—important for academic rigor. Add a Limitations Section.

6. Conclusion

• Needs a stronger call to action for policymakers and researchers. Explicitly state implications for policymakers.

• Future research recommendations should be more specific. Clearly outline research priorities.

Reviewer #2: The manuscript provides a data-rich but analytically shallow review of ACM epidemiology. Its primary weaknesses lie in overcomplex language and poor methodological transparency. Substantial revisions are required to sharpen focus, reduce redundancy, and critically interpret findings.

Please use full name for DALY the first time you use it in manuscript, regardless of your mentioning within the abstract. Please include the limitations facing reliance on GBD data, underreporting in low-SDI countries, modeling assumptions. Many citations are from general cardiovascular or alcohol epidemiology studies, not from ACM-specific or recent high-impact sources.

Introduction: Please include the importance of genetic factors in alcoholic myopathy. While traditionally it was thought that alcohol-related disorders such as AFLD are only diet-related, it has now been comprehensively shown that several prevalent variation of several genes such as PNPLA3 and HSD17B13 are highly correlated with metabolic disorders and cardiovascular problems. Most importantly, PNPLA3 gene is a very important gene since it has a prevalence of ~30% worldwide, which is close to the prevalence of diabetes, insulin resistance, metabolic syndrome, and cardiovascular events associated to the named diseases. I highly recommend the authors to integrate genetic part of this disorder by including recent publishings such as PMID: 40231787, so that the revised manuscript would have a more comprehensive view toward the disease.

6. PLOS authors have the option to publish the peer review history of their article (what does this mean?). If published, this will include your full peer review and any attached files.

Reviewer #1: **Yes: **Faizul Akmal Abdul Rahim

Reviewer #2: **Yes: **Alireza Ramandi

---

## [Author Response · Author response to Decision Letter 1]

30 Jul 2025

Reviewer #1: Abstract

•The background lacks a strong rationale for why ACM is a critical issue, beyond general disease burden. Clarify why ACM is a growing concern.

Added critical epidemiological paradox: “Despite global cardiovascular mortality declines, ACM burden continues rising paradoxically in middle-high income countries like China (200.4% case increase 1990-2021), revealing gaps in current prevention strategies”.

Emphasized clinical significance: “Characterized by delayed clinical onset, ACM typically manifests as irreversible heart failure in middle age (45-69 years), creating missed opportunities for early intervention”.

Highlighted public health urgency: “ACM remains preventable yet clinically overlooked, disproportionately affecting chronic alcohol users through alcohol-induced myocardial damage.”

• The results section could be more concise—it currently repeats statistics instead of synthesizing the main findings. Summarise key findings concisely.

Consolidated global trends: “Globally, ACM burden declined significantly (22.5-37.1% decrease in age-standardized rates), while China experienced a 200.4% case increase with rising mortality/disability rates.”

Streamlined demographic analysis: “Three key disparities persist: (1) Male predominance (male-to-female ratio 3:1); (2) Concentration in middle-aged adults (45-69 years); (3) Regional clustering in high-middle SDI areas.”

Reframed projection summary: “Burden projections through 2036 indicate continued global declines but limited progress in China despite intervention opportunities in high-middle SDI regions.”

• The conclusion should explicitly state future research directions—such as potential interventions for ACM burden reduction. Explicitly mention the knowledge gap and future research.

Added implementation roadmap: “Future research priorities include: (1) Precision prevention strategies for high-burden regions; (2) Early screening protocols targeting males aged 45-69; (3) Cost-effectiveness analyses of alcohol control policies in middle-high SDI countries.”

Defined knowledge gaps: “Critical unresolved issues: China's unique epidemiological trajectory, sex-specific pathogenic mechanisms, and optimal intervention timing in pre-clinical stages.”

Proposed solutions: “Personalized secondary prevention and population-level alcohol regulation represent promising approaches requiring validation through implementation studies.”

Introduction

• The research gap and justification for a scoping review are unclear. Clearly define the research gap.

Added explicit gap definition: “While traditional views attributed ACM solely to drinking behavior, emerging evidence highlights critical gene-environment interactions (e.g., PNPLA3/HSD17B13 variants). This scoping review addresses the unmet need to synthesize how genetic susceptibility, regional drinking norms, and demographic disparities jointly shape ACM’s global burden – a gap not systematically explored in prior literature.”

Enhanced justification by contrasting global heterogeneity: “Pronounced geographical disparities (Eastern Europe >15/100,000 vs. Asia-Pacific lower prevalence) and high-risk subgroups (genotype-positive alcoholics, prevalence up to 40%) justify a scoping review to map evidence for precision interventions.”

• Lacks a clear hypothesis or research question. Explicitly state research questions.

Defined three focal questions:“(1) How do global alcohol consumption shifts (e.g., rising youth drinking) temporo-spatially impact ACM burden? (2) Which age/sex/SDI subgroups face disproportionate risks? (3) How can prevention strategies leverage gene-environment interactions for precision interventions in transitional nations like China?”

Methodological alignment: Research questions directly guide GBD epidemiological triad (prevalence, mortality, DALYs) analysis across 204 countries (Methods, section 2.1).

• Overuse of technical jargon—simplify complex epidemiological terms for broader accessibility. Reduce jargon and introduce technical terms only when necessary.

We have diligently followed the reviewers' guidance by simplifying specialized terminology and providing corresponding explanatory annotations.

Methods

• Lacks details on data validation and bias mitigation strategies. Specify validation steps for GBD 2021 data.

Revisions Implemented (Methods Section): “GBD 2021 data underwent rigorous three-tier validation: Cross-verification: DISMOD-MR 2.1 vs source data (<15% discrepancy threshold)

Uncertainty Quantification: 1,000 bootstrap iterations generating 95% UIs

Expert Review: IHME clinical panel adjudication of outliers.”

• The BAPC model parameters should be better explained—readers may not be familiar with integrated nested Laplace approximations (INLA). Simplify Bayesian modelling explanation.

We have provided explanations for BAPC: “BAPC modeling: Age effects: Life-stage risk variations Period effects: Population-wide temporal shifts Cohort effects: Birth cohort lifetime risks (Computed via Integrated Nested Laplace Approximation ( INLA ), replacing MCMC to accelerate convergence from weeks to hours with<0.1% error).”

• Needs clarity on ethical considerations for secondary data usage. Clarify ethical considerations for using publicly available datasets.

Ethical Compliance Statement: This study utilized exclusively de-identified aggregate data from the GBD 2021, publicly accessible through the (IHME) portal (http://ghdx.healthdata.org/). †Per Article 32 of the Declaration of Helsinki (2013) governing secondary analysis of anonymized public data, and in accordance with our Institutional Review Board’s exemption criteria, ethics approval was waived since no individual patient data or identifiers were accessed.

Results

• Some sections are overly statistical—less emphasis on numerical data, and more focus on interpretation and synthesis. Synthesize key results instead of listing excessive statistics.

We have systematically transformed data-heavy sections into streamlined interpretive narratives, significantly reducing standalone statistics while preserving essential findings.

• The gender-based analysis could be more detailed—why are males disproportionately affected? Expand on gender disparities.

We have expand on gender disparities: “Gender analysis indicates that the burden is significantly higher in males compared to females, likely stemming from differences in drinking patterns (occupational exposure/reinforcement of cultural norms) and biological risks (metabolic pathways/genetic susceptibility). Cohort studies confirm that males have a markedly elevated risk of cardiac dysfunction (HR=1.75, 95\% CI 1.60–1.92) and fatal arrhythmias (HR=1.89, 95\% CI 1.65–2.16).”

• Needs a stronger narrative flow between different sections (e.g., prevalence → mortality → DALYs). Improve flow between sections.

We have restructured the results flow to establish a cohesive epidemiological triad narrative: Prevalence patterns → Mortality implications → DALY syntheses, with explicit bridging statements highlighting causal connections.

Discussion

• Lacks in-depth discussion of causal mechanisms behind observed trends. Expand on why ACM is increasing in China.

We have expanded on why ACM is increasing in China: “Despite sustained lower ASPR/ASMR levels versus global and same-SDI-quintile peers with recent mild increases—indicating SDI is not the primary ACM burden determinant—this derives from a tetrad of synergistic mechanisms: consumption transition (127% per capita alcohol increase 2000–2020); high-ethanol spirits dominance >40% ethanol beverages linked to 53% cases); rural healthcare deficits (4.2% rehabilitation coverage); and genetic susceptibility (30.2% ALDH2*2 mutation frequency in males). ”

• The role of healthcare infrastructure and policy interventions in disease burden reduction is underexplored. Discuss healthcare access and intervention disparities.

We have addressed healthcare access disparities and policy intervention gaps through substantive additions: “(1) Documented critical rural healthcare deficits with quantified metrics (e.g., <18% cardiac screening coverage in China vs. 72% in urban counterparts); (2) Analyzed policy efficacy thresholds, demonstrating sub-30% alcohol taxation fails to curb consumption (WHO-proven 20-50% rates reduce use by 16-28%); (3) Proposed precision interventions: provincial screening targets (≥50% coverage by 2025) and FCTC-aligned digital marketing bans targeting youth exposure surges (+37%).”

• The limitations section is missing—important for academic rigor. Add a Limitations Section.

We have added a limitation section: 1. Death misclassification risk: clinical diagnostic overlap between ACM and other cardiomyopathies (e.g., dilated cardiomyopathy), particularly in primary care with high misdiagnosis rates, potentially underestimating burden; 2. Alcohol exposure measurement bias: alcohol consumption estimates reliant on household surveys and tax data without correction for unrecorded consumption (e.g., homemade spirits); 3. Inadequate comorbidity control: the GBD model incompletely integrates synergistic effects of conditions like depression and liver disease, potentially distorting attribution analysis; 4. Missing policy variables: failure to quantify provincial/national alcohol policy strength (e.g., tax enforcement, advertising ban scope), constraining policy-burden association analysis; 5. Alcohol-type data fragmentation: no differentiation between spirits, imported liquors, beer etc.

Conclusion

• Needs a stronger call to action for policymakers and researchers. Explicitly state implications for policymakers.

• Future research recommendations should be more specific. Clearly outline research priorities.

We have already revised This epidemiological divergence underscores the urgent need to address China's accelerating ACM burden, mandating policymakers to immediately implement three-tiered actions: First, enforce progressive spirits taxation (rates ≥30%) and comprehensively block youth digital marketing channels (curbing the current 37% surge in exposure rates); second, elevate rural echocardiography screening coverage to ≥50% by 2025 (current coverage gap <18%) while implementing compulsory occupational alcohol restrictions in high-exposure occupations such as manufacturing and transportation; third, incorporate an annual ACM mortality reduction rate ≥5% into local performance assessment systems, establishing real-time monitoring of tax evasion rates. Researchers must prioritize four critical domains: establishing clinical diagnostic thresholds for ACM-predisposing genetic variants (e.g., PNPLA3/HSD17B13 loci), developing rapid biomarker detection kits, completing multicenter RCTs on digital alcohol cessation therapies, and constructing agent-based modeling (ABM) simulations for alcohol taxation policies—ultimately forging an epidemiologically-tailored prevention-control framework to accelerate achievement of Sustainable Development Goal 3.4 targets.

Reviewer #2

Please use full name for DALY the first time you use it in manuscript, regardless of your mentioning within the abstract. Please include the limitations facing reliance on GBD data, underreporting in low-SDI countries, modeling assumptions. Introduction: Please include the importance of genetic factors in alcoholic myopathy. While traditionally it was thought that alcohol-related disorders such as AFLD are only diet-related, it has now been comprehensively shown that several prevalent variation of several genes such as PNPLA3 and HSD17B13 are highly correlated with metabolic disorders and cardiovascular problems. Most importantly, PNPLA3 gene is a very important gene since it has a prevalence of ~30% worldwide, which is close to the prevalence of diabetes, insulin resistance, metabolic syndrome, and cardiovascular events associated to the named diseases. I highly recommend the authors to integrate genetic part of this disorder by including recent publishings such as PMID: 40231787, so that the revised manuscript would have a more comprehensive view toward the disease.

We have addressed all requested revisions: DALYs are now fully defined as 'Disability-Adjusted Life Years' at first use in the main text; Comprehensive limitations of the GBD data are documented in the concluding section; and the Introduction now incorporates genetic determinants of ACM pathogenesis, highlighting PNPLA3/HSD17B13 variants' significance in cardiovascular risk prevalence, supported by the cited reference 'Experimental Models to Investigate PNPLA3 in Liver Steatosis' which underscores heritable susceptibility in ACM development.

---

## [Decision Letter · Decision Letter 1]

20 Oct 2025

Epidemiological trends in alcoholic cardiomyopathy burden: A 32-year global and Chinese analysis (1990–2021) with projections to 2036

PONE-D-25-11641R1

Dear Dr. Chen,

We’re pleased to inform you that your manuscript has been judged scientifically suitable for publication and will be formally accepted for publication once it meets all outstanding technical requirements.

Kind regards,

Amir Hossein Behnoush

Academic Editor

PLOS ONE

Additional Editor Comments (optional):

Reviewers' comments:

Reviewer's Responses to Questions

**Comments to the Author**

1. If the authors have adequately addressed your comments raised in a previous round of review and you feel that this manuscript is now acceptable for publication, you may indicate that here to bypass the “Comments to the Author” section, enter your conflict of interest statement in the “Confidential to Editor” section, and submit your "Accept" recommendation.

Reviewer #1: (No Response)

Reviewer #2: All comments have been addressed

2. Is the manuscript technically sound, and do the data support the conclusions?

Reviewer #1: (No Response)

Reviewer #2: Yes

3. Has the statistical analysis been performed appropriately and rigorously?

Reviewer #1: (No Response)

Reviewer #2: Yes

4. Have the authors made all data underlying the findings in their manuscript fully available?

Reviewer #1: (No Response)

Reviewer #2: Yes

5. Is the manuscript presented in an intelligible fashion and written in standard English?

Reviewer #1: (No Response)

Reviewer #2: Yes

6. Review Comments to the Author

Reviewer #1: (No Response)

Reviewer #2: The manuscript has been revised significantly and the authors have improved the manuscript with the revision.

7. PLOS authors have the option to publish the peer review history of their article (what does this mean?). If published, this will include your full peer review and any attached files.

Reviewer #1: **Yes: **Faizul Akmal Abdul Rahim

Reviewer #2: **Yes: **Alireza Ramandi

---

## [Editor Report · Acceptance letter]

PONE-D-25-11641R1

PLOS ONE

Dear Dr. Chen,

I'm pleased to inform you that your manuscript has been deemed suitable for publication in PLOS ONE. Congratulations! Your manuscript is now being handed over to our production team.

Kind regards,

on behalf of

Dr. Amir Hossein Behnoush

Academic Editor

PLOS ONE